# Oh-A-Dino: Understanding and Enhancing Attribute-Level Information in Self-Supervised Object-Centric Representations

## Abstract

Object-centric understanding is fundamental to human vision and required for complex reasoning. Traditional methods define slot-based bottlenecks to learn object properties explicitly, while recent self-supervised vision models like DINO have shown emergent object understanding. We investigate the effectiveness of self-supervised representations from models such as CLIP, DINOv2 and DINOv3, as well as slot-based approaches, for multi-object instance retrieval, where specific objects must be faithfully identified in a scene. This scenario is increasingly relevant as pre-trained representations are deployed in downstream tasks, e.g., retrieval, manipulation, and goal-conditioned policies that demand fine-grained object understanding. Our findings reveal that although self-supervised and slot-based representations faithfully encode geometric attributes (shape, size), for non-geometric attributes (colours, material, texture), the embedding structure does not appear to reflect these attributes in a way that geometry-based similarity measures (e.g., cosine distance) can access, limiting their usefulness for object-level reasoning tasks. We show that learning an auxiliary latent space over segmented patches, where VAE regularisation enforces compact, disentangled object-centric representations, recovers this embedding structure. Augmenting the self-supervised methods with such latents improves retrieval across all attributes, suggesting a promising direction for making self-supervised representations more reliable in downstream tasks that require precise object-level reasoning.

## 1 Introduction

Object-centric understanding is central to human vision and a prerequisite for complex reasoning (van Steenkiste et al., 2019; Schölkopf et al., 2021). Learning such representations remains challenging: slot-based methods explicitly disentangle object properties (Locatello et al., 2020; Wu et al., 2023), but may compromise global scene understanding (Montero et al., 2024), while large self-supervised models such as DINO and DINOv2 (Caron et al., 2021; Oquab et al., 2023) show emergent object structure without task-specific supervision. This makes them attractive as general-purpose vision backbones, increasingly used in downstream tasks that require object-level reasoning (Uelwer et al., 2023; Seitzer et al., 2023; Zhou et al., 2024; Wagner & Harmeling, 2024; Szot et al., 2023).

Many of these downstream tasks such as retrieval, embodied manipulation, or goal-conditioned policies depend on representations that can reliably capture the properties needed to disambiguate between multiple objects in a scene. Whether current pre-trained and object-centric representations provide this level of fidelity, however, remains unclear.

We ask: *Do existing representations encode the attribute-relevant information needed to reliably disambiguate objects in multi-object settings?*

Our findings reveal that while self-supervised vision models and slot-based representations excel at capturing geometric structure such as shape and size, surface-level cues such as colour and material do not appear to be encoded in the representation geometry in a way that geometry-based similarity measures (e.g., cosine distance) can access. This leads to systematic errors when disambiguating otherwise similar objects. To address this, we propose *Object-Aware-DINO (Oh-A-DINO)* which augments self-supervised representations from DINOv2-S with object-centric latent vectors learned

from segmented patches using a Variational Autoencoder (Kingma et al., 2019) (VAE). The VAE's regularisation encourages a compact latent space that captures these fine-grained properties, and concatenating the latents with global features improves alignment across all object attributes.

We demonstrate that this approach improves multi-object instance retrieval on CLEVR (Johnson et al., 2017) and CLEVRTex (Karazija et al., 2021), with particularly notable gains in colour and material matching. We also show that these improvements transfer to real-world instance retrieval settings on the Stanford Cars (Krause et al., 2013) dataset. Overall, this highlights a broader challenge: widely used pre-trained representations, though powerful, may miss the kinds of attribute-level information that downstream tasks need for accurate object-level reasoning.

**Our contributions are threefold:** (i) We identify a key limitation of both self-supervised and slot-based representations: they capture geometric attributes well but do not encode non-geometric object attributes such as colour and material in a metrically accessible way, which is essential for distinguishing objects in multi-object scenes in fully unsupervised settings where no labels or decoders are available.

(ii) We propose a simple and modular method to augment self-supervised representations with object-centric latents learned from a VAE, without retraining the backbone.

(iii) We demonstrate that this approach improves retrieval across all attributes, providing more reliable object-level representations and suggesting a promising direction for enhancing pre-trained representations in downstream tasks that require detailed object reasoning.

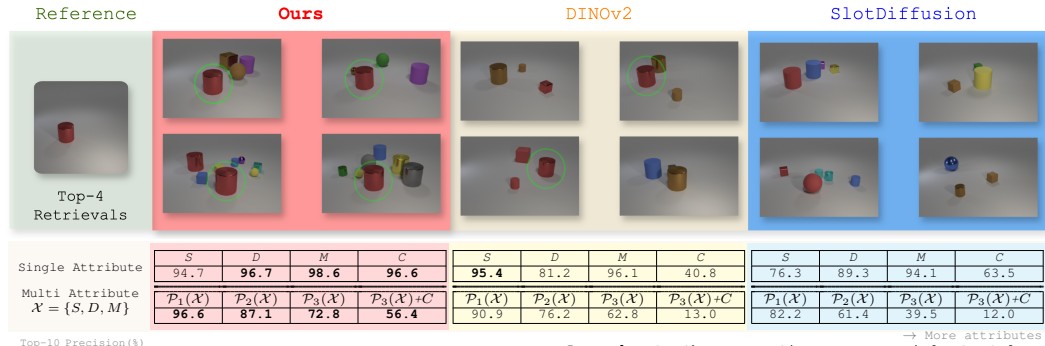

Figure 1: **DINOv2 representations and slot-based representations struggle at multi-object instance retrieval, due to weak object-specific features (largely colour) in its embeddings. Our method is able to retrieve more relevant images by combining general scene and local representations.** DINOv2 representations excel at retrieving images where multiple attributes match such as shape and size, while retrieval degrades when adding colour. Slot-based representations lack specialisation, where retrievals sometimes match fine-grained features such as colour but often failing to retrieve a similar object altogether. Our method performs the best being able to augment the DINOv2 representation to mitigate its shortcomings.

## 2 ANALYSING REPRESENTATION QUALITY WITH OBJECT-CENTRIC INSTANCE RETRIEVAL

To evaluate how well existing methods capture object-centric detail, we compare DINOv2 (Caron et al., 2021), DINOv3 (Siméoni et al., 2025), CLIP (Radford et al., 2021), SlotDiffusion (Wu et al., 2023), SPOT (Kakogeorgiou et al., 2024), and SmoothSA (Zhao et al., 2025) representations in multi-object instance retrieval. We use the CLEVR and CLEVRTex datasets, which provide attribute labels (shape, size, material, colour), enabling us to probe whether these properties are encoded and accessible in the learned embeddings. Beyond single attributes, we also test combinations, since downstream applications often require correctly binding multiple attributes to the same object.

Figure 1 illustrates the setup: the top row shows qualitative examples of top-4 retrievals for a single reference object, while the bottom row reports top-10 precision across individual attributes and

Table 1: **Object-level features improve prediction for all attributes, while also improving retrieval of multiple simultaneous attributes.** The main improvement is achieved on the colour (CLEVR) and material (CLEVRTex) attribute, which indicates that our method mitigates for the inability to access surface-level information in the other methods' representations. Slot-based methods perform worse than the pretrained DINOv2 representations when predicting multiple attributes simultaneously.

| | Attrib. Ablation Top-10 Precision (%) | CLEVR | | | | CLEVRTex | | | CLEVR | | | CLEVRTex | |
|---|---|---|---|---|---|---|---|---|---|---|---|---|---|
| | | Shape | Size | Mat. | Col. | Shape | Size | Mat. | $\mathcal{P}_2(\mathcal{X})$ | $\mathcal{P}_3(\mathcal{X})$ | $\mathcal{P}_3(\mathcal{X})+C$ | $\mathcal{P}_2(\mathcal{X})$ | $\mathcal{P}_2(\mathcal{X})+C$ |
| | Oh-A-DINOv2 **(Ours)** | 94.7±2.0 | **96.7**±2.2 | **98.6**±0.0 | **96.6**±2.6 | **90.7**±2.1 | 85.3±1.5 | **45.8**±3.9 | **85.2**±2.1 | **72.8**±1.4 | **56.4**±1.6 | **56.0**±0.2 | **20.3**±2.6 |
| SSL | DINOv2 | 95.4±1.1 | 81.2±1.2 | 96.1±1.0 | 40.8±2.1 | 87.3±3.8 | 80.3±2.6 | 24.2±1.9 | 76.2±3.1 | 62.8±1.2 | 13.0±1.1 | 54.0±0.1 | 12.2±2.1 |
| | DINOv3 | **96.9**±1.3 | 85.5±1.5 | 97.0±1.2 | 44.5±1.9 | 81.3±0.7 | 86.8±1.2 | 11.8±1.1 | 80.1±1.2 | 64.8±0.6 | 13.7±1.2 | 44.5±0.2 | 4.2±1.5 |
| | CLIP | 83.8±2.3 | 83.2±1.8 | 98.3±0.2 | 66.8±1.7 | 81.4±1.1 | **90.1**±1.4 | 15.0±1.1 | 63.0±0.8 | 40.9±0.9 | 14.6±1.5 | 43.9±0.3 | 3.1±0.9 |
| OCL | SlotDiff. | 76.3±1.2 | 89.3±1.3 | 94.1±1.5 | 63.5±0.4 | 67.0±1.7 | 80.2±1.6 | 6.6±0.9 | 56.3±0.4 | 39.5±1.3 | 12.0±1.2 | 33.9±2.2 | 1.4±0.4 |
| | SPOT | 94.5±1.4 | 86.6±1.8 | 94.2±0.8 | 39.7±1.7 | 77.1±1.5 | 79.9±0.5 | 9.1±1.0 | 73.5±0.7 | 53.7±0.6 | 10.1±0.8 | 41.3±0.5 | 2.2±0.4 |
| | SmoothSA | 90.8±1.6 | 85.7±2.3 | 95.2±0.6 | 38.6±1.5 | 72.4±1.5 | 77.7±1.9 | 7.6±1.2 | 70.6±0.7 | 50.1±0.2 | 8.7±1.5 | 35.5±1.6 | 1.7±0.6 |

Single-Attribute Retrieval      Multi-Attribute Retrieval

attribute subsets. Formally, we compute means over $\mathcal{P}_i(\mathcal{X}) = \{A \in \mathcal{P}(\mathcal{X}) : |A| = i\}$, where $\mathcal{X} = \{\text{shape, size, material, colour}\}$, and extend the evaluation to ordered permutations. We further present extended results in Table 1.

**Colour is the weakest attribute across models.** DINOv2 and DINOv3 both perform strongly on shape (95.4%, 96.9%) but fall sharply on colour (40.8%, 44.5%). This indicates that newer self-supervised objectives have not fully addressed colour encoding. CLIP improves colour substantially (66.8%), but at the cost of weaker shape and size performance compared to DINO models (83.8% shape, 83.2% size). SlotDiffusion also retrieves colour better than DINOv2 (63.5% vs. 40.8%), yet underperforms on shape (76.3%), suggesting a trade-off between geometric and surface-level features.

**Slot-based methods struggle with attribute binding.** In multi-attribute retrieval, SlotDiffusion degrades rapidly: while colour can be matched in isolation, it fails to consistently bind multiple attributes to the same object. Visual inspection confirms that while DINOv2 and DINOv3 retrievals often fail only by colour mismatch, SlotDiffusion frequently retrieves objects that share little resemblance with the query. Furthermore, methods such as SPOT and SmoothSA which exhibit improved object segmentation improve in geometric feature retrieval, but barely improve or worsen on colour and texture(material).

**CLIP shares the same geometric bias despite different supervision.** CLIP shows relatively strong performance on single attributes such as material (98.3%), but its precision collapses in the CLEVR multi-attribute setting, especially when colour is included (14.6%). This is noteworthy because, unlike self-supervised models trained with heavy colour jitter, CLIP is not explicitly trained to be colour-invariant. Nevertheless, it exhibits the *same* pattern of relying primarily on geometry-derived cues while underrepresenting fine-grained surface attributes.

Overall, these results reveal a consistent pattern: self-supervised and slot-based models capture geometric structure well, but surface-level cues such as colour and material are often encoded in a way that is not metrically accessible. This likely stems from their inductive biases and bottlenecks, which prioritise features predictive of object identity while downweighting appearance cues that vary across otherwise similar objects. Consequently, colour and material may exist in the representation but cannot be recovered through similarity-based operations used in many downstream tasks. This motivates our approach: instead of modifying pre-training, we introduce a lightweight object-centric latent space. Using DINOv2-S as the backbone, we train a VAE on segmented patches and augment the global DINO features with these latents, enabling the model to encode surface-level attributes more directly in the representation.

## 3 ENHANCING SELF-SUPERVISED REPRESENTATIONS WITH OBJECT-CENTRIC LATENTS

Our approach, illustrated in Figure 2, enriches self-supervised features with object-level latents to better capture fine-grained attributes. We will show that DINOv2 features benefit from this the most.

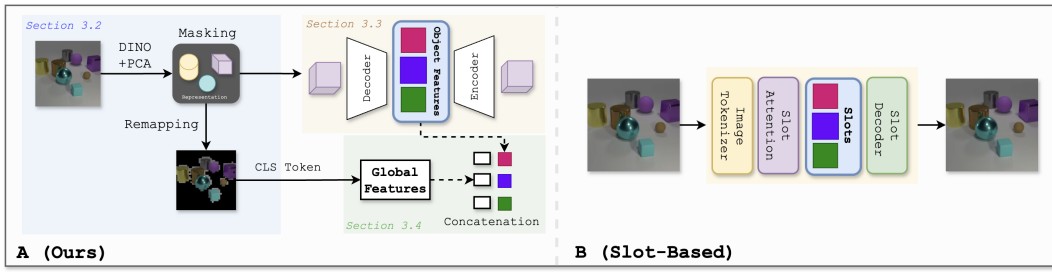

Figure 2: **Our method leverages the implicit general object understanding of DINOv2 to extract object-level features for multi-object instance retrieval. B:** Traditional slot-based object-centric methods learn slot-representations via cross-attention which provide little inductive bias, while having to compress global and object-level features into a small set of latents. **A:** We propose combining general self-supervised features with learnt object-level features to obtain an improved latent representation. The object-level features are learnt from image patches making training efficient and the latent space expressive.

We proceed in four steps: (i) extract patch embeddings from DINOv2, (ii) segment objects using PCA, (iii) learn a latent space over object patches with a VAE, and (iv) combine global and local features into a joint representation.

## 3.1 PRELIMINARIES

Given an image $x \in \mathbb{R}^{h \times w \times c}$, we extract patch embeddings $y \in \mathbb{R}^{p \times n_y}$ from a pre-trained DINOv2 model, where $p$ is the number of patches. These embeddings form the basis for both segmentation and global features.

## 3.2 EXTRACTING OBJECT-LEVEL FEATURES WITH PCA

We extract object-level patches from DINOv2 embeddings using a simple PCA-based segmentation procedure (Figure 2). While DINOv3 can be used for this segmentation procedure as well, in practice we found DINOv2 to produce better segmentation for our use case. This involves three steps: (i) separating foreground from background, (ii) refining object consistency, and (iii) remapping the mask to the original image to obtain patches. We describe the process in the following along with Figure 3.

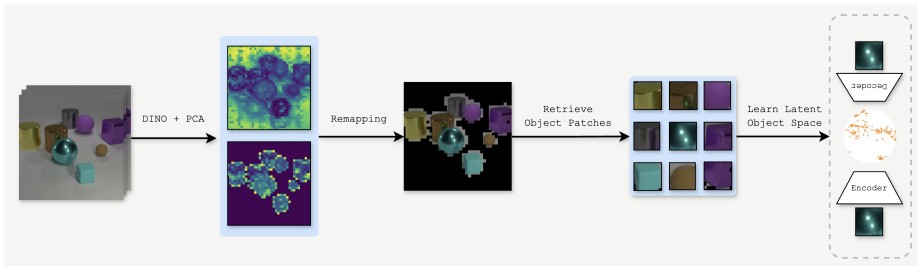

Figure 3: **Detailed view of extracting object-level features using PCA.** From $t$ images we create a segmentation mask which is used to extract the object image patches. We then learn a latent space of the object patches with a VAE.

**(i) Foreground–background separation.** We first flatten all patch embeddings $Y \in \mathbb{R}^{t \cdot p \times n_y}$ collected from the batch of $t$ images and apply PCA. To stabilise segmentation, we compute PCA over a batch of $t = 50$ images. We elaborate on how to collect this batch in Appendix B.1. The first principal component $Z' \in \mathbb{R}^{t \cdot p}$ reliably distinguishes background from foreground content, consistent with prior findings on DINO features (Siméoni et al., 2021; Wang et al., 2023). A binary mask is obtained by thresholding against the median:

$$\hat{M}^{\text{fg}} = \mathbb{1}(Z' > \text{median}(Z')). \tag{1}$$

This mask selects patches belonging to salient objects.

**(ii) Refining object consistency.** While this initial mask captures saliency, it can fragment objects. We therefore reapply PCA on the foreground embeddings $Y^{\text{fg}} = \hat{M}^{\text{fg}} \odot Y$, yielding a refined mask $M^{\text{fg}}$. This second pass improves spatial coherence, avoiding masks that cut across single objects.

**(iii) Remapping to image space.** Finally, we map the binary mask of patches back to the original image resolution. Each mask entry corresponds to a patch of size $s_h \times s_w$; we expand these back into image coordinates, producing a segmented image $x^o$ and a set of object patches

$$\rho = \{\rho_1, \rho_2, \ldots, \rho_p\}, \quad \rho_i \in \mathbb{R}^{s_h \times s_w \times c}. \tag{2}$$

These patches serve two purposes: (a) they provide a segmented image $x^o$ from which we later extract global features, and (b) they form the training input for the VAE in Section 3.3.

### 3.3 LEARNING OBJECT-LEVEL LATENT REPRESENTATIONS WITH A VAE

To encode fine-grained object attributes, we train a Variational Autoencoder on the segmented patches. The encoder maps each patch $\rho$ to a latent vector $z \in \mathbb{R}^{n_z}$, while the decoder reconstructs the patch. The VAE loss is

$$\mathcal{L}_{\text{VAE}} = \mathbb{E}_{z \sim q_\phi(z|\rho)}[\log f_\theta(\rho|z)] - \beta D_{\text{KL}}(q_\phi(z|\rho) \, \| \, \mathcal{N}(0, 1)), \tag{3}$$

with a small $\beta$ to encourage attribute-specific clustering. This yields a set of object-level latents $z \in \mathbb{R}^{p \times n_z}$, which will be combined with SSL representations in the following step.

### 3.4 COMBINING GLOBAL AND LOCAL REPRESENTATIONS

We combine the global scene feature from the self-supervised model (CLS token in DINO's case) $\epsilon \in \mathbb{R}^{n_g}$ with the object-level latents. For each patch, we concatenate $\epsilon$ with its VAE latent to form

$$v = [\epsilon, z] \in \mathbb{R}^{p \times (n_g + n_z)}. \tag{4}$$

This structured representation retains global context while injecting object-level detail. Retrieval is then performed by cosine similarity between $v$ and $v'$ from query and candidate images as shown in Appendix A

## 4 RELATED WORK: OBJECT-CENTRIC LEARNING AND INSTANCE RETRIEVAL

**Slot-based object centric learning.** Recent advances in object-centric representation learning (e.g., SlotAttention (Locatello et al., 2020), MONet (Burgess et al., 2019), SlotDiffusion (Wu et al., 2023), AdaSlot (Fan et al., 2024), Slot-VAE (Wang et al., 2023)) have enabled models to decompose images into object-level features, making it possible to reason about individual objects within a scene. These methods mostly focus on learning disentangled representations of individual objects by enforcing an object-level slot-based bottleneck via cross attention, which then allows for faithful image reconstruction, generation of novel object configurations, or object segmentation. In order to extract the objects, the image is usually tokenised and an attention mechanism is applied to the tokens (e.g. SLATE (Singh et al., 2021)). A slot decoder then reconstructs the original image from the slot representation. STEVE (Singh et al., 2022) reconstructs the image from the slot-representation autoregressively to learn dependencies between different image patches i.e. tokens. More recent methods, have further exploited this by attempting to incorporate global scene level features into the representations (Chen et al., 2024) or by adding compositional reasoning into the slot-based representations (Jung et al., 2024).

**Combining self-supervised features with task-specific features.** Recent methods such as SLATE (Singh et al., 2021) and STEVE (Singh et al., 2022) have leveraged vision transformers for object-centric learning to learn improved slot representations that also capture global scene information via attention mechanisms. These methods have shown improved performance on object-centric tasks, but they require retraining the complete model, which can be computationally expensive and time-consuming. Similarly, DINOSAUR (Seitzer et al., 2023) and GOLD (Chen et al., 2024) use DINO representations to provide better natural priors to learn object-centric representations on

natural images. R$^2$Former (Zhu et al., 2023) learns global correspondences between patches from a transformer pre-trained on ImageNet and then leverages local attention maps to rerank retrieved patches of candidate images. Zhang et al.(Zhang et al., 2023) combine learnt Stable Diffusion and DINOv2 features to enhance instance retrieval. We build on this work by combining representations that occupy different semantic spaces, unlike the previously studied Stable Diffusion and DINOv2 features which share similar characteristics. This integration of semantically distinct vectors (i.e. general features and object-level features) enhances our model's capabilities.

**Assessing the quality of representations with instance retrieval.** Instance retrieval is a challenging task that requires models to retrieve images with similar object attributes and configurations. Traditionally, this task has been best solved by methods that leverage representations from pre-trained models, such as ResNet (He et al., 2016) or vision transformers (Dosovitskiy et al., 2020). Generally, instance retrieval depends on learning a good latent space or representation of the image, where retrievals are ranked on a similarity measure (Chen et al., 2021). Naturally, instance retrieval has been used to evaluate the quality of representations in self-supervised models (Caron et al., 2021; Oquab et al., 2023). Since we are interested in evaluating the quality of the representations for multi-object understanding, we choose the task of multi-object instance retrieval. Beyond traditional vision applications, multi-object instance retrieval plays a crucial role in goal-conditioned reinforcement learning (GCRL) and robotic tasks (Zheng et al., 2024). In GCRL, agents often rely on goal images to specify desired configurations of objects in the environment, particularly in tasks requiring manipulation or navigation in multi-object settings (Le Paine et al., 2019). Similar challenges are present in hierarchical reinforcement learning (HRL), where sub-goals often involve interacting with specific object sets. Robust multi-object retrieval can inform option selection and improve learning efficiency (Hafner et al., 2022). Moreover, the emergence of embodied agents has seen the need for pre-trained representations that understand nuance in their environment (Szot et al., 2023; Szot et al., 2024).

## 5 EXPERIMENTS

A key challenge in evaluating object-centric representations is finding datasets that expose systematic variation in object attributes while remaining relevant for real-world tasks. Most natural-image benchmarks lack consistent multi-object scenes with fine-grained labels, while purely synthetic datasets may be dismissed as too controlled. We therefore combine synthetic datasets for diagnostic evaluation with a real-world dataset for external validation.

### 5.1 EXPERIMENTAL SETUP

**Retrieval tasks.** We evaluate representations on a retrieval task where the goal is to recover objects with matching attributes given a reference image. For CLEVR and CLEVRTex, attribute labels allow systematic evaluation of **shape**, **size**, **material**, and **colour** (CLEVR) or **shape**, **size**, and **material** (CLEVRTex, with material combining texture and colour). For Stanford Cars, which lacks attribute annotations, we focus on appearance-level cues by computing the average deviation in RGB pixel values to assess colour consistency. Dataset splits are reported in Appendix C, Table 7.

**Part I: Synthetic diagnostic benchmarks.** On CLEVR and CLEVRTex, we carry out both (i) *performance evaluation* and (ii) *attribute-level analysis*. First, we measure retrieval quality using Top-10 precision, weighted precision, and error rate across all attribute subsets, probing how well each model preserves object identity. Second, we perform an ablation study which was presented in Table 1 to examine retrieval fidelity for individual attributes and combinations, highlighting failures in binding surface-level cues (e.g. colour, texture). In this context, we analyse the effect of using standalone VAE features and show whether VAE-augmented variants of DINOv2, DINOv3, CLIP, and SlotDiffusion improve attribute alignment. Finally, we test whether increasing backbone capacity (DINOv2-L) helps encode surface-level features.

**Part II: Real-world benchmark.** To validate beyond synthetic settings, we repeat our retrieval experiments on the Stanford Cars dataset. Here, cars of the same class often share nearly identical geometry but differ in colour and surface details, providing a natural testbed for appearance-sensitive retrieval. This complements the controlled synthetic benchmarks and demonstrates that the limitations

we identify are not confined to toy datasets. In particular, we measure the *average Euclidean distance in colour space* between the query and retrieved instances.

For training, we fit the VAE on 2,000 CLEVR/CLEVRTex images segmented into object patches. Evaluation is performed on 500 query images and 5,000 candidates for retrieval, with Stanford Cars used only for evaluation under its standard splits.

**Evaluation metrics.** We report three metrics:

(i) *Top-10 Precision:* fraction of the first ten retrieved images that match the reference attributes;

(ii) *Weighted Precision:* same as above, but weighted by retrieval rank ($1/i$ for rank $i$), emphasising higher-ranked matches;

(iii) *Error Rate:* percentage of queries for which no correct retrieval is found in the top ten (for CLEVRTex, normalised for queries with fewer than ten valid candidates).

**Baselines.** For each method we run seven trials with 50 query images and 5,000 candidate images, sampling queries without replacement. SlotDiffusion is pre-trained separately on CLEVR and CLEVRTex. Seeds are chosen at random and kept consistent across models. We report standard deviation across runs.

## 5.2 CLEVR AND CLEVRTEX: MULTI-OBJECT INSTANCE RETRIEVAL

Table 2: **Oh-A-DINOv2 improves over SSL and slot-based features for multi-object instance retrieval.** Our method outperforms both self-supervised features and slot-based approaches. The table shows that Oh-A-DINOv2 achieves higher precision and lower error rates on CLEVR and CLEVRTex, indicating that combining DINOv2 with object-level VAE features addresses the limitations of pure SSL features and slot-attention methods.

|  | Main Results (%) | CLEVR | | | CLEVRTex | | |
|---|---|---|---|---|---|---|---|
|  |  | Top-10 Precision↑ | Weighted Precision↑ | Error Rate↓ | Top-10 Precision↑ | Weighted Precision↑ | Error Rate↓ |
|  | Oh-A-DINOv2 (Ours) | **56.4**±2.0 | **49.0**±2.2 | **1.0**±0.0 | **20.3**±2.6 | **11.2**±1.4 | **41.0**±1.0 |
| SSL | DINOv2 | 13.0±1.1 | 13.8±1.2 | 28.0±1.0 | 12.2±2.1 | 8.0±1.3 | 55.6±0.7 |
| SSL | DINOv3 | 13.7±1.2 | 14.5±1.3 | 28.0±2.0 | 4.2±1.1 | 1.5±0.3 | 82.4±1.9 |
| SSL | CLIP | 14.7±1.5 | 16.3±1.6 | 29.2±1.9 | 2.9±1.2 | 0.7±0.2 | 90.6±0.4 |
| OCL | SlotDiff. | 12.0±1.2 | 12.7±1.3 | 33.6±1.5 | 1.4±0.4 | 0.9±0.2 | 93.4±0.2 |
| OCL | SPOT | 10.1±1.8 | 11.0±1.8 | 37.7±1.0 | 2.2±0.4 | 1.2±0.2 | 85.7±0.3 |
| OCL | SmoothSA | 8.7±1.5 | 9.5±1.7 | 42.6±0.6 | 1.7±0.6 | 1.2±0.3 | 87.4±0.2 |

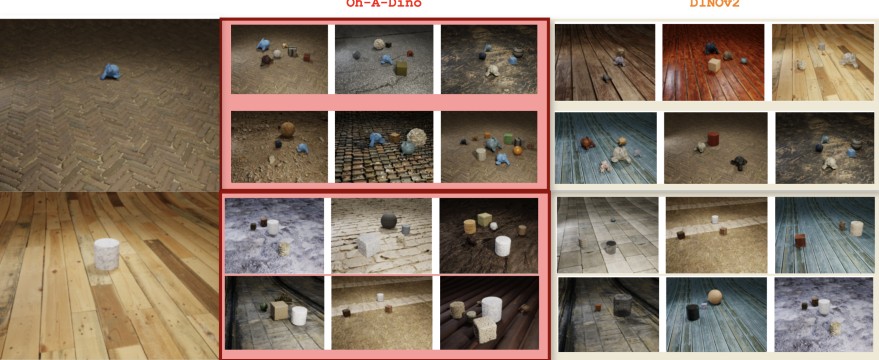

Figure 4: **Visualisations of retrievals for CLEVRTex, augmenting with VAE object level features allows for retrieval in complex multi-object scenes.** Oh-A-Dino consistently retrieves the object in question with the correct properties such as the shape, size and material. DINOv2 on the other hand often retrieves the correct object but with the wrong texture. Further visualisations, also for other methods can be found in Appendix E.

**Our approach improves multi-object instance retrieval.** We show in Table 2 that the learnt object features are expressive and complement the DINOv2 representations well. On CLEVR, Oh-A-

DINOv2 achieves a Top-10 Precision of **56.4%**, compared to only 13.0% for DINOv2, 13.7% for DINOv3, 14.7% for CLIP, 12.0% for SlotDiffusion, 10.1% for SPOT, and 8.7% for SmoothSA. On CLEVRTex, Oh-A-DINOv2 reaches **20.3%**, substantially higher than the 12.2% of DINOv2, the 4.2% of DINOv3, the 2.9% of CLIP, and the 1.4% of SlotDiffusion. Interestingly, DINOv3 performs markedly worse than DINOv2 on CLEVRTex (4.2% vs. 12.2%), suggesting that the newer model is less robust to textured multi-object scenes. These results together with Table 1 confirm that SSL features (DINO/CLIP) fail to capture non-geometric attributes, slot-based features collapse under increased scene complexity, and that combining SSL (specifically DINOv2) with object-level VAE representations directly addresses both shortcomings.

### 5.2.1 THE ROLE OF THE VAE FEATURES

Table 3: **Object-level VAE features enhance color and multi-attribute retrieval.** While the standalone VAE improves fine-grained color retrieval on both CLEVR and CLEVRTex, only Oh-A-DINOv2 effectively integrates this signal with DINOv2 scene-level features. This leads to substantial gains in multi-attribute retrieval, where pure VAE features or SSL features (DINOv2) alone fall short.

| VAE Features (%) | CLEVR | | | CLEVRTex | | |
|---|---|---|---|---|---|---|
| | Top-10 Precision↑ | Weighted Precision↑ | Error Rate↓ | Top-10 Precision↑ | Weighted Precision↑ | Error Rate↓ |
| VAE (Ours) | $53.1_{\pm3.6}$ | $29.2_{\pm2.7}$ | $2.0_{\pm1.1}$ | $11.7_{\pm3.7}$ | $4.1_{\pm1.3}$ | $60.0_{\pm3.4}$ |
| Oh-A-DINOv2 (Ours) | $\mathbf{56.4}_{\pm2.0}$ | $\mathbf{49.0}_{\pm2.2}$ | $\mathbf{1.0}_{\pm0.0}$ | $\mathbf{20.3}_{\pm2.6}$ | $\mathbf{11.2}_{\pm1.4}$ | $\mathbf{41.0}_{\pm1.0}$ |
| DINOv2 | $13.0_{\pm1.1}$ | $13.8_{\pm1.2}$ | $28.0_{\pm1.0}$ | $12.2_{\pm2.1}$ | $8.0_{\pm1.3}$ | $55.6_{\pm0.7}$ |

| Attrib. Ablation Top-10 Precision (%) | CLEVR | | | | CLEVRTex | | | CLEVR | | | CLEVRTex | |
|---|---|---|---|---|---|---|---|---|---|---|---|---|
| | Shape | Size | Mat. | Col. | Shape | Size | Mat. | $\mathcal{P}_2(\mathcal{X})$ | $\mathcal{P}_3(\mathcal{X})$ | $\mathcal{P}_3(\mathcal{X})+C$ | $\mathcal{P}_2(\mathcal{X})$ | $\mathcal{P}_2(\mathcal{X})+C$ |
| VAE (Ours) | $93.4_{\pm1.2}$ | $\mathbf{99.2}_{\pm0.4}$ | $\mathbf{99.6}_{\pm0.3}$ | $\mathbf{98.7}_{\pm0.4}$ | $89.0_{\pm0.9}$ | $\mathbf{94.4}_{\pm0.8}$ | $43.5_{\pm5.3}$ | $82.8_{\pm0.5}$ | $63.3_{\pm0.5}$ | $53.1_{\pm3.6}$ | $52.5_{\pm0.1}$ | $11.7_{\pm3.7}$ |
| Oh-A-DINOv2 (Ours) | $94.7_{\pm2.0}$ | $96.7_{\pm2.2}$ | $98.6_{\pm0.0}$ | $96.6_{\pm2.6}$ | $\mathbf{90.7}_{\pm2.1}$ | $85.3_{\pm1.5}$ | $\mathbf{45.8}_{\pm3.9}$ | $\mathbf{85.2}_{\pm2.1}$ | $\mathbf{72.8}_{\pm1.4}$ | $\mathbf{56.4}_{\pm1.6}$ | $\mathbf{56.0}_{\pm0.2}$ | $\mathbf{20.3}_{\pm2.6}$ |
| DINOv2 | $\mathbf{95.4}_{\pm1.1}$ | $81.2_{\pm1.2}$ | $96.1_{\pm1.0}$ | $40.8_{\pm2.1}$ | $87.3_{\pm3.8}$ | $80.3_{\pm2.6}$ | $24.2_{\pm1.9}$ | $76.2_{\pm3.1}$ | $62.8_{\pm1.2}$ | $13.0_{\pm1.1}$ | $54.0_{\pm0.1}$ | $12.2_{\pm2.1}$ |

Single-Attribute            Multi-Attribute

**Object-level features improve multi-object instance retrieval when combined with DINOv2.** Table 3 (bottom) shows that individual attribute prediction improves significantly with the learned object-level representations VAE for both CLEVR and CLEVRTex. This improves color prediction in Oh-A-DINOv2. However, Table 3 (top) reveals that only the Oh-A-DINO combination manages to utilise this color signal to improve multi-attribute retrieval. When comparing Oh-A-DINOv2 and VAE in CLEVR, even with a smaller performance gap in CLEVR Top-10 precision, our method delivers substantially better weighted precision, indicating that DINO's scene features help retrieve more consistent objects. This is further validated in Appendix E.3, where we show that Oh-A-DINO consistently performs the closest retrievals regardless of exact matching.

Table 4: **Ablation on VAE augmentation and model capacity.** (left) Augmenting SSL and slot-based features with object-level VAE latents improves retrieval across methods, with the largest gains observed for Oh-A-DINOv2. (right) Scaling backbone capacity from DINOv2-S to DINOv2-L provides only minor improvements.

| Augmenting with VAE (%) | CLEVR | | | CLEVRTex | | |
|---|---|---|---|---|---|---|
| | Top-10 Precision↑ | Weighted Precision↑ | Error Rate↓ | Top-10 Precision↑ | Weighted Precision↑ | Error Rate↓ |
| Oh-A-DINOv2 (Ours) | $\mathbf{56.4}_{\pm2.0}$ | $\mathbf{49.0}_{\pm2.2}$ | $\mathbf{1.0}_{\pm0.0}$ | $\mathbf{20.3}_{\pm2.6}$ | $\mathbf{11.2}_{\pm1.4}$ | $\mathbf{41.0}_{\pm1.0}$ |
| DINOv3-VAE | $34.8_{\pm2.6}$ | $25.2_{\pm1.9}$ | $10.8_{\pm1.8}$ | $5.7_{\pm1.6}$ | $1.0_{\pm0.2}$ | $82.0_{\pm1.8}$ |
| CLIP-VAE | $55.0_{\pm2.4}$ | $31.8_{\pm1.5}$ | $1.0_{\pm0.0}$ | $3.1_{\pm1.0}$ | $0.9_{\pm0.2}$ | $86.4_{\pm1.2}$ |
| SlotDiff.-VAE | $14.3_{\pm1.5}$ | $6.8_{\pm0.6}$ | $30.6_{\pm2.0}$ | $2.8_{\pm0.1}$ | $0.8_{\pm0.0}$ | $87.2_{\pm1.7}$ |

| Model Capacity (%) | CLEVR | | | CLEVRTex | | |
|---|---|---|---|---|---|---|
| | Top-10 Precision↑ | Weighted Precision↑ | Error Rate↓ | Top-10 Precision↑ | Weighted Precision↑ | Error Rate↓ |
| Oh-A-DINOv2 (Ours) | $\mathbf{56.4}_{\pm2.0}$ | $\mathbf{49.0}_{\pm2.2}$ | $\mathbf{1.0}_{\pm0.0}$ | $\mathbf{20.3}_{\pm2.6}$ | $\mathbf{11.2}_{\pm1.4}$ | $\mathbf{41.0}_{\pm1.0}$ |
| DINOv2-S | $13.0_{\pm1.1}$ | $13.8_{\pm1.2}$ | $28.0_{\pm1.0}$ | $12.2_{\pm2.1}$ | $8.0_{\pm1.3}$ | $55.6_{\pm0.7}$ |
| DINOv2-L | $13.3_{\pm1.2}$ | $14.8_{\pm1.1}$ | $24.0_{\pm2.0}$ | $13.2_{\pm1.8}$ | $9.0_{\pm1.1}$ | $50.6_{\pm0.9}$ |

### 5.2.2 AUGMENTING OTHER SSL MODELS WITH VAE LATENT FEATURES

**Augmenting SSL features with VAE latents is effective, but depends on the backbone.** Table 4 (left) further compares augmentations of different SSL and slot-based features with our learnt VAE latents. For DINOv3-VAE, improvements on CLEVR are moderate (34.8% Top-10 Precision) and largely vanish on CLEVRTex. In contrast, CLIP-VAE shows strong gains on CLEVR (55.0% Top-10 Precision, comparable to Oh-A-DINOv2) but only minor improvements on CLEVRTex. Finally, combining SlotDiffusion with either DINOv2 or VAE provides negligible benefit, suggesting that

slot-based representations are not sufficiently disentangled to compose well with VAE latents. Overall, these results highlight that while VAE augmentation consistently helps SSL features, the degree of improvement depends strongly on the underlying representation, with DINOv2 benefiting the most.

### 5.2.3 EFFECT OF MODEL CAPACITY

**Scaling up SSL model capacity does not resolve the retrieval bottleneck.** Table 4(right) compares DINOv2-S with its larger variant DINOv2-L. While DINOv2-L achieves a slight improvement over DINOv2-S on both CLEVR (Top-10 Precision 13.3% vs. 13.0%) and CLEVRTex (13.2% vs. 12.2%), the gains are marginal and far from closing the gap to Oh-A-DINOv2 (**56.4%** on CLEVR, **20.3%** on CLEVRTex). This suggests that simply increasing model capacity within SSL frameworks does not lead to more disentangled or attribute-sensitive object representations. By contrast, augmenting with object-level VAE features directly addresses the compositional limitations of SSL features, providing consistent improvements irrespective of backbone size.

### 5.3 STANFORD CARS: REAL-WORLD INSTANCE RETRIEVAL

| Main Results (%) | Stanford Cars |
| --- | --- |
| | Distance to Ref Colour↓ |
| Oh-A-DINOv2 (Ours) | **0.432**$_{\pm 0.011}$ |
| DINOv2 | 0.512$_{\pm 0.014}$ |
| CLIP | 0.524$_{\pm 0.021}$ |

Figure 5: **Our method improves over baselines and leverages DINOv2's general object understanding.** (left) Table: comparison against baselines on Stanford Cars shows that our method achieves lower distance to the colours of the reference image. (right) Oh-A-Dino can consistently retrieve cars which match orientation and shape while matching colour and material opposed to DINOv2. Further real-world examples can be found in Section E.1

**VAE features help retrieve surface level attributes more faithfully for Stanford Cars.** Figure 5 (left) shows that Oh-A-DINOv2 achieves a lower distance to the reference colour (**0.432**) compared to both DINOv2 (0.512) and CLIP (0.524). This indicates that Oh-A-DINOv2 not only captures the global shape and orientation of cars but is also more sensitive to fine-grained attributes such as colour and material, which are crucial for reliable retrieval in real-world settings. The qualitative examples on the right confirm this behaviour: our method consistently retrieves cars that align in orientation and shape while matching the query's visual appearance. We provide further qualitative analysis on ImageNet (Deng et al., 2009) samples in the Appendix in Section E.1

## 6 DISCUSSION

**Limitations and Future Work.** Our method relies on obtaining reasonable object segmentations to extract patch-level latents; in complex scenes this may be more difficult, though we use PCA on DINO features only for simplicity and any segmentation method could be substituted. The quality of the pre-trained backbone also matters: when surface-level cues are weak or entangled, the resulting latents may be ambiguous or dominated by local patches, though in practice we found the global DINO features and VAE latents to balance well. Finally, while our approach captures colour and material robustly, nuanced textures remain challenging, as seen in CLEVRTex, suggesting room for future work on learning richer surface-level representations. Broader analysis of why SSL models downweight appearance cues, and evaluation on real-world control settings where subtle attribute differences matter, would further strengthen these findings.

**Conclusion.** We showed that self-supervised and slot-based models struggle to make surface-level attributes accessible in their representations. A lightweight VAE trained on segmented patches complements DINOv2 by encoding these cues directly, improving multi-object retrieval on synthetic and real data. This suggests that augmenting pre-trained models with simple object-centric latents is a promising path toward more reliable object-level reasoning.

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

# 7  APPENDIX

## A  CALCULATING SIMILARITIES.

We calculate the similarity on patch level between a query image $x$ and a candidate image $x'$ using the representations $v$ and $v'$. Since $v$ and $v'$ share the same embedding dimension, we can calculate the similarity between the two patches $v_i \in \mathbb{R}^{n_v}$ and $v'_j \in \mathbb{R}^{n_v}$ by computing the pairwise cosine similarity between the two representations:

$$S_{ij} = \frac{v_i \cdot v'_j}{||v_i|| \cdot ||v'_j||} \in \mathbb{R}, \tag{5}$$

for all $i, j \in \{1, \ldots, p\}$.

Then, for each query patch $v_i$ we retrieve the patch with the highest cosine similarity, i.e., the most similar patch from $v'$

$$s_i^{\max} = \max_{j=\{1,\ldots,p\}} S_{ij}. \tag{6}$$

The list $S^{\max} = [s_1^{\max}, \ldots, s_p^{\max}]$ then contains the cosine similarities for the most similar patches from $v'$ for each patch in $v$. Finally, our *similarity score* of the query image $x$ with another image $x'$ is the average of the similarities in $S^{\max}$, that is $\bar{s} = \text{average}(S^{\max})$. Note, if we compare $x$ against $n$ candidate images $x'$, we will have $n$ similarity scores for the given query image $x$.

## B  LEARNING OBJECT-LEVEL FEATURES WITH PCA

### B.1  CREATING BATCHES TO SABILISE PCA

Our PCA segmentation requires computing the first principal component of a set of patch embeddings. For this we need to retrieve a batch of $t$ images that help stabilise PCA.

Concretely, we maintain an offline bank of DINOv2 patch embeddings extracted from 500 randomly sampled training images. For a new image, we then first retrieve the $t - 1$ nearest neighbours, which corresponds to batch size $t$ with the sample in question added. We then perform PCA on the union of the neighbour patches and the test image's patches, yielding a stable estimate of the foreground–background direction.

This also allows us to apply PCA segmentation using only a single test image at inference time. Importantly, this step is modular: any method that provides coarse foreground–background separation (e.g. SAM, OCL masks) can be substituted without modifying the remainder of the pipeline.

### B.2  THE BENEFIT OF REAPPLYING PCA

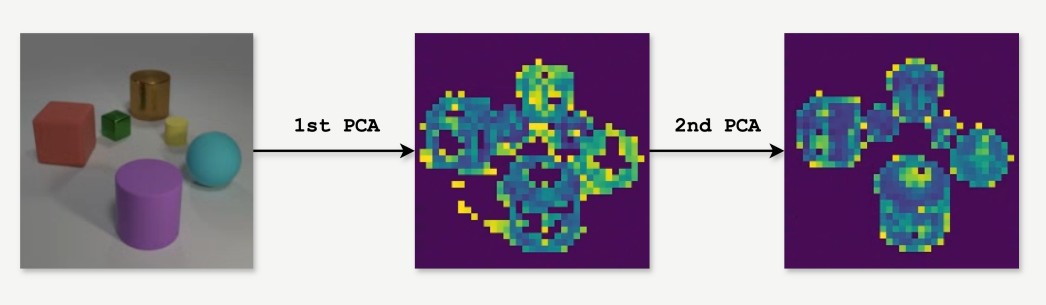

Figure 6: We show the benefit of reapplying PCA to the foreground object-level features. The first PCA passthrough to separate the background and foreground features, while the second PCA pass-through helps to separate the individual objects and get a continuous mask across the objects.

In Figure 6, we show the benefit of applying PCA a second time, i.e., over the segmented foreground features after separating background from foreground. While the first PCA provides a decent mask,

some noisy patches from the background still remain and some patches on the objects are segmented away. Applying PCA a second time on the foreground patch features which were separated from the background in the first pass, helps in retrieving a better overall segmentation mask.

## B.3 SEGMENTATION EXAMPLES

We how examples of segmentations for CLEVR, CLEVRTex, Stanford Cars and ImageNet in Figure 7. We show that segmenetation works well by applying PCA to the DINO features, segmenting the relevant objects in the scene. We also see that the segmentations do not have to be pixel perfect for our method to work, since the VAE learns the distribution of surface-level properties from the patches.

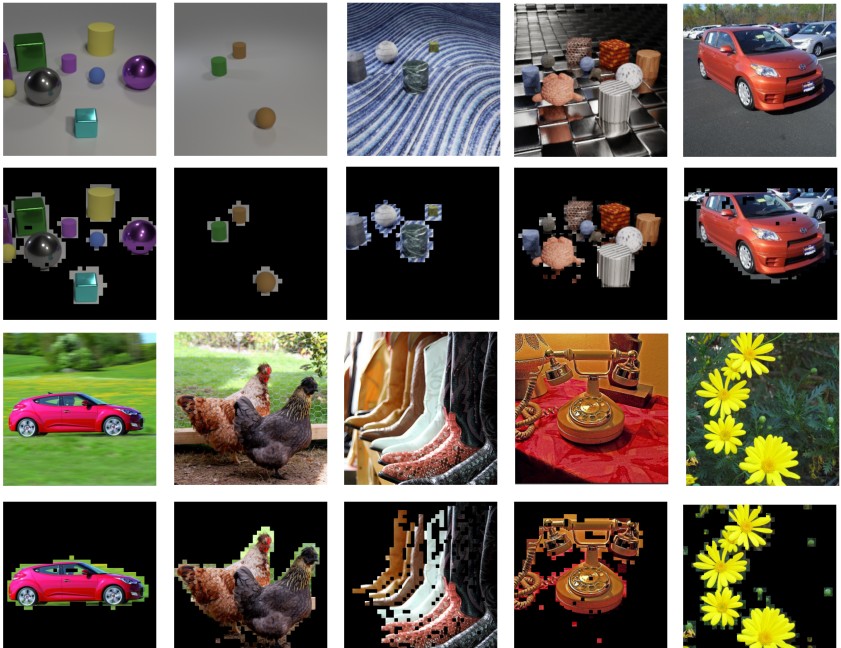

Figure 7: **Segmentation with PCA of DINO features works well, but segmentation masks do not have to be pixel perfect.**

## C ADDITIONAL DATASET DETAILS

### C.1 DATASET STATISTICS

| Dataset | #Images | #Objects | #Shapes | #Colors | #Materials | #Backgrounds |
|---------|---------|----------|---------|---------|------------|--------------|
| **CLEVR** | 100k | 3-10 | 3 | 8 | 2 | 1 |
| **CLEVRTex** | 50k+10k | 3-10 | 4 | – | 60 | 60 |

Table 5: Dataset characteristics for CLEVR and CLEVRTex. The CLEVRTex dataset has additional textures for the objects and backgrounds compared to CLEVR. It also has no color information for the objects, which is replaced by the material information. In general, instance retrieval with the CLEVRTex dataset is much harder due to the increased number of possible object configurations.

We provide details on the number of attributes and objects in the CLEVR and CLEVRTex dataset in Table 5. While both datasets have the same number of possible objects in an image, the CLEVRTex Dataset has different materials compared to the CLEVR dataset which has 8 colors and 2 materials. This presents different challenges to the learnt reprsentations. In the CLEVR case, the representations must account for redundancy in the object features, while for the CLEVRTex dataset the large number

of materials makes retrieval more challenging since some materials are quite similar and only differ slightly.

## C.2   TRAINING-TEST SET SIZES

**CLEVR/CLEVRTex Dataset Splits**

| Train | Validation-Query | Candidates | Query Size |
|-------|------------------|------------|------------|
| 2000  | 500              | 5000       | 50         |

Table 6: **Data splits for the CLEVR and CLEVRTex datasets.** The train set is used to train the VAE, while the validation set is used to select query images for the retrieval task. The test set contains the candidate images for the retrieval task.

**Stanford Cars Dataset Splits**

| Validation-Query | Candidates | Query Size |
|------------------|------------|------------|
| 175              | 5000       | 25         |

Table 7: **Data splits for Stanford Cars dataset.** We only evaluate on the Stanford Cars Dataset where we take a subset of 5000 candidate images and 175 query images split into groups of 25.

## D   EXPERIMENT DETAILS

### D.1   VAE HYPERPARAMETERS

**VAE Hyperparameters**

| Parameter | Value |
|-----------|-------|
| Input size | $64 \times 64 \times 3$ |
| Latent size | 32 |
| Learning rate | $1e-4$ |
| $\beta$ | $1e-4$ |

### D.2   DINO HYPERPARAMETERS

**DINO Hyperparameters**

| Parameter | Value |
|-----------|-------|
| Input size | $518 \times 518$ |
| Embedding Dimension | 384 |
| Patch Size | 14 |

Note that for DINOv2 and DINOv3 we used the non register variants since we noticed no difference in our analysis and the non register variants were faster.

### D.3   COMPUTE

The experiments where performed on an RTX 3090 24GB VRAM with Ryzen 3900X CPU. The main bottleneck is running the experiments which requires caching all the required query and candidate images. For this, we used 128GB of RAM.

## E   ADDITIONAL VISUALISATIONS

### E.1   ADDITIONAL REAL WORLD RESULTS: IMAGENET

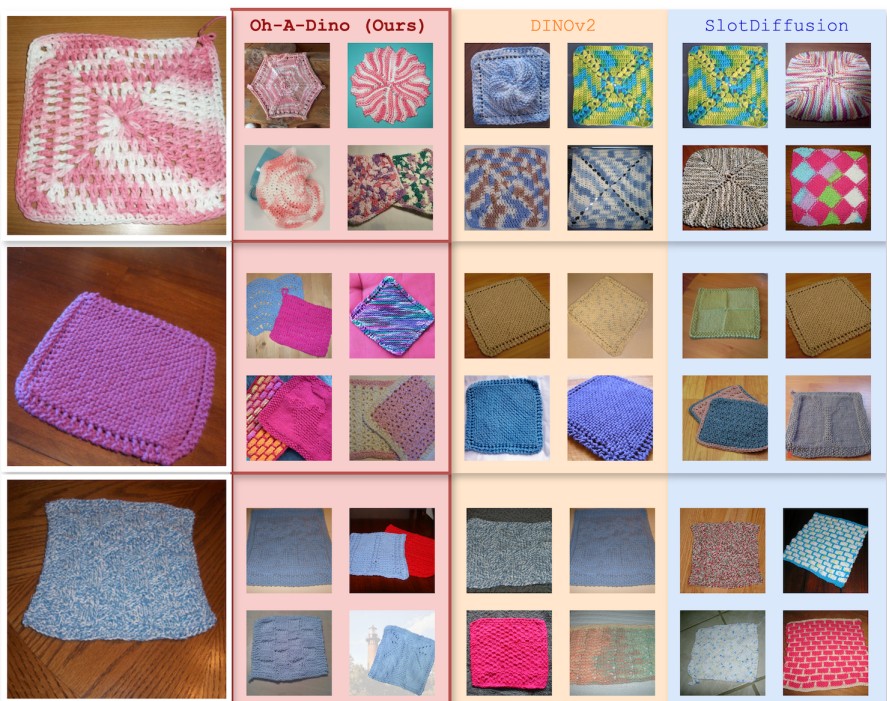

Figure 8: **Our method is effective at retrieval with ImageNet examples being able to retrieve better matches that stay faithful to the object's material while matching multiple colours. In the example in row one, we see that even mixes of colours can be retrieved successfully.**

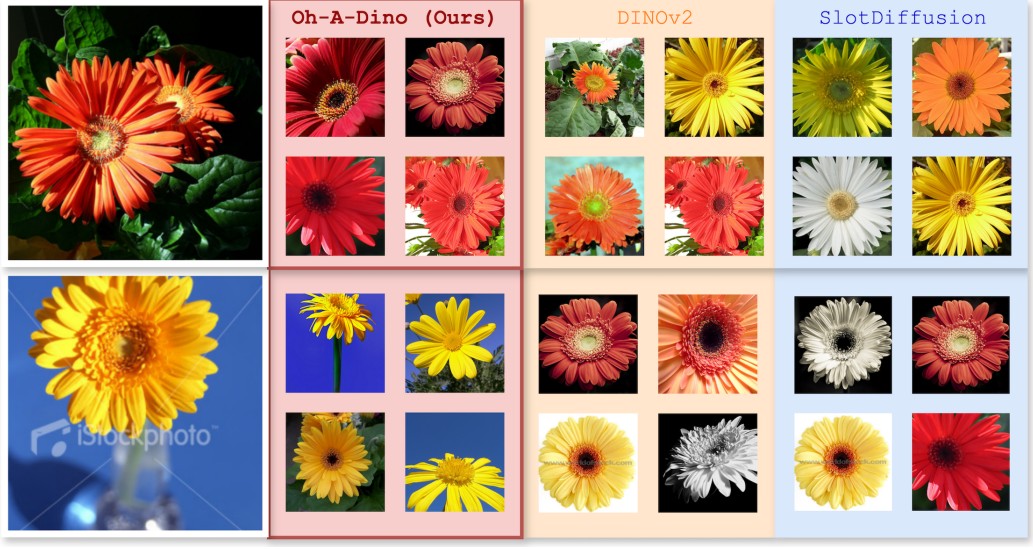

Figure 9: **Flowers in the ImageNet datset are complex and exhibit different lighting conditions. Our method is arguably able to retrieve colour and match lightning more precisely than DINOv2 and SlotDiffusion, retrieving the correct colour for all four retrievals. Other methods manage to retrieve similar colours or other colours and lighting conditions (see black and white) altogether.**

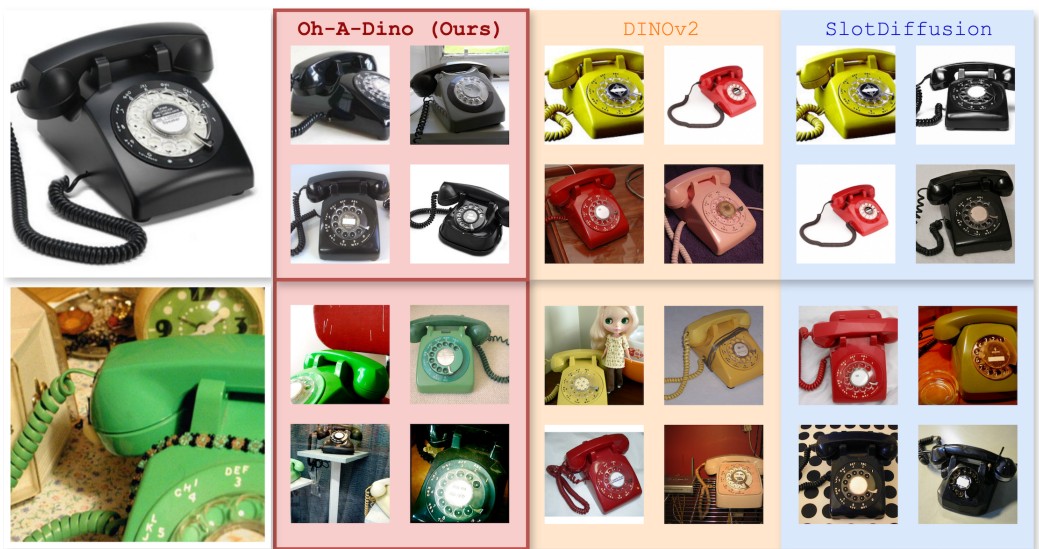

Figure 10: **Objects with complex geometries both in the foreground and background are not an issue either as we are able to retrieve the correct colour phones while staying faithful to orientation, material and colour. In row one, we are able to retrieve 4 correct black colour phones, while other methods mix different colours. This validates our findings from Table 1, where our method not only improves material and colour retrieval, but also positively complements the geometric attributes.**

Across a diverse set of ImageNet examples including textiles, flowers, and consumer objects, our method consistently retrieves instances that preserve fine-grained surface attributes such as colour, material, and lighting, while maintaining geometric consistency. In Figure 8, textile objects with mixed or subtle colour patterns are matched faithfully, even when multiple hues appear within a single object. Flower images (Figure 9), which exhibit substantial variation in illumination, background clutter, and viewpoint, are retrieved with correct colour and comparable lighting conditions, outperforming both DINOv2 and SlotDiffusion. Finally, for more complex objects such as mobile phones (Figure 10), our approach successfully identifies matches that align not only in colour but also in material appearance and orientation, demonstrating that enriching the representation with object-masked VAE latents complements rather than disrupts the geometric structure of the backbone. These findings reinforce the quantitative improvements reported earlier: our method reliably makes surface-level attributes geometrically accessible without sacrificing performance on geometric cues.

## E.2 MULTI-OBJECT AND CLUTTERED BACKGROUNDS

Figure 11 illustrates retrieval results on more complex ImageNet samples that reflect the reviewer's concern. In contrast to earlier examples, these queries involve richly cluttered scenes with multiple co-occurring objects, varying illumination, and substantial background distraction. Despite these challenges, our method continues to identify candidates that closely preserve the object's surface-level attributes, including subtle color and material cues, even when these are partially occluded or surrounded by unrelated distractors. These examples reinforce that the advantages of our approach extend beyond simple or canonical ImageNet instances and hold under realistic, highly complex conditions.

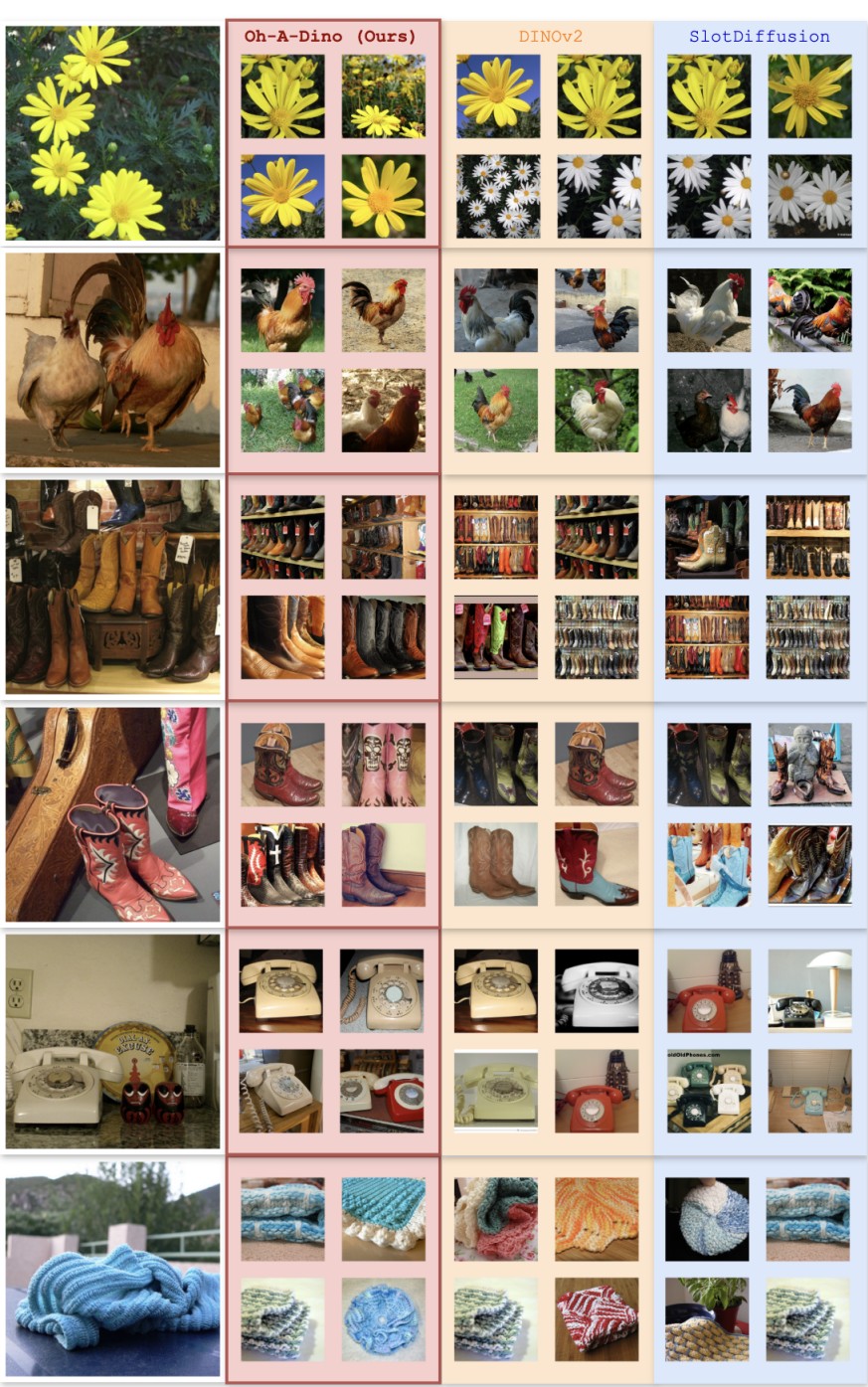

Figure 11: **Additional ImageNet Retrieval Examples under Complex Real-World Conditions: we present challenging ImageNet queries that include multiple objects, cluttered and heterogeneous backgrounds, occlusions, and high intra-class variation. Our method consistently retrieves matches that resemble the object's color, material, and fine-grained appearance attributes despite significant scene complexity.**

### E.3 CLEVRTEX

We show 3 retrievals from the query dataset for CLEVRTex where relative performance is comparable, i.e., the retrievals are ranked similar in terms of retrieval performance within each model. We see that Oh-A-DINO, performs the best retrievals, in most cases being able to retrieve the exact object or an object similar to the reference object. DINOv2 as presented in Figure 1 retrieves the correct object, however with the wrong texture properties. For the VAE retrievals, we see that the object properties are understood correctly, however with no real object attribute binding, often retrieving the wrong objects or the background in the material of the object. This shows, that the combination of the VAE representation an the DINOv2 representation allows for correctly interpreting the background while performing consistent object retrieval. SlotDiffusion performs the worst, being very sensitive to the background. Even when masking the background with our approach, results barely improved.

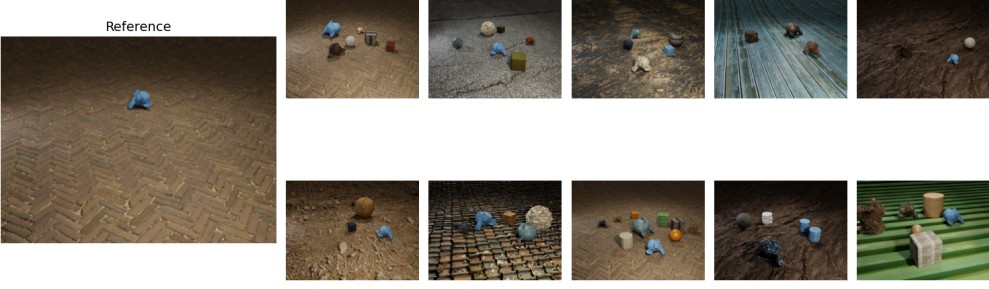

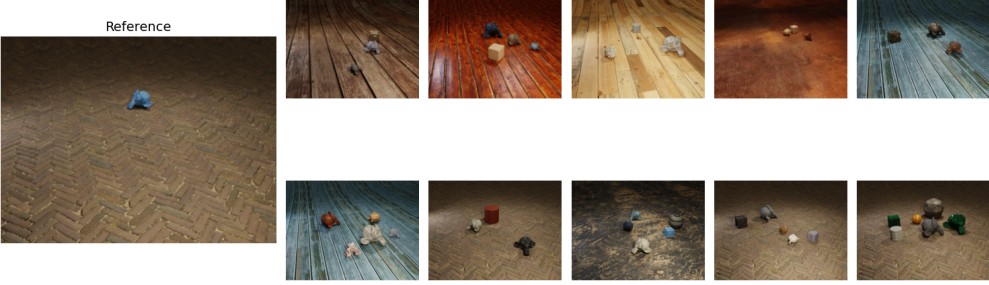

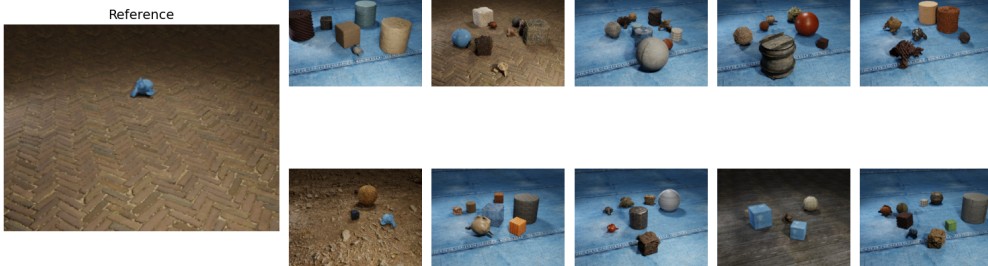

Top-10 Retrievals with SlotDiffusion

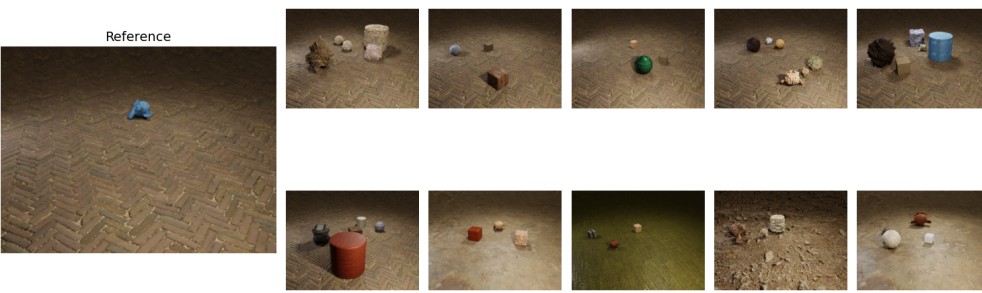

Top-10 Retrievals with Oh-A-DINO

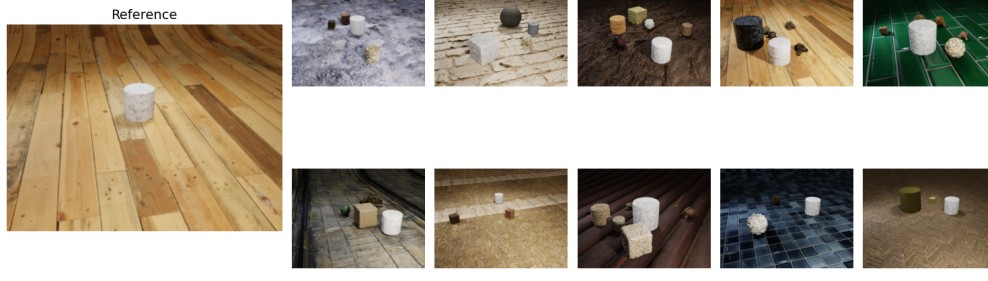

Top-10 Retrievals with DINOv2

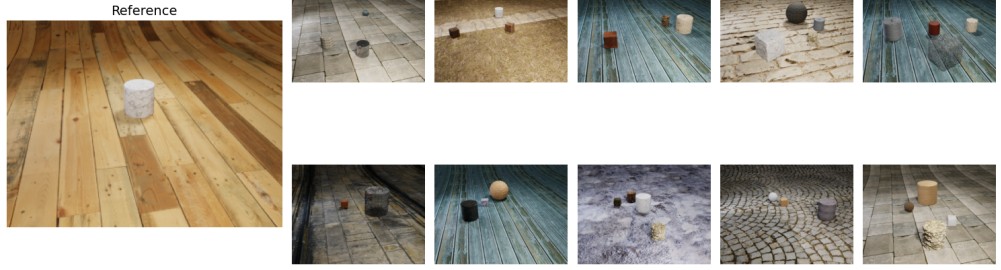

Top-10 Retrievals with VAE

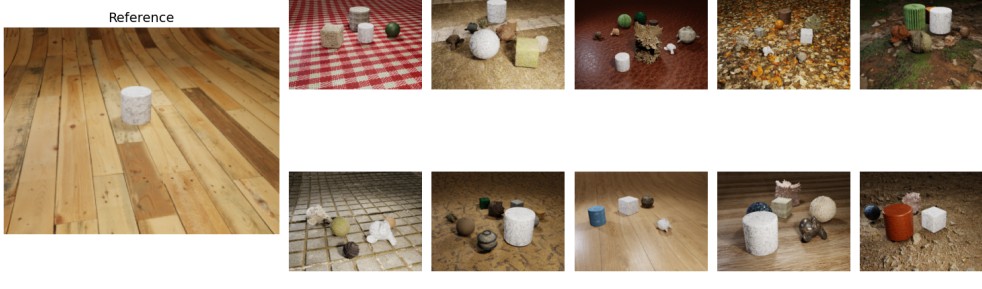

Top-10 Retrievals with SlotDiffusion

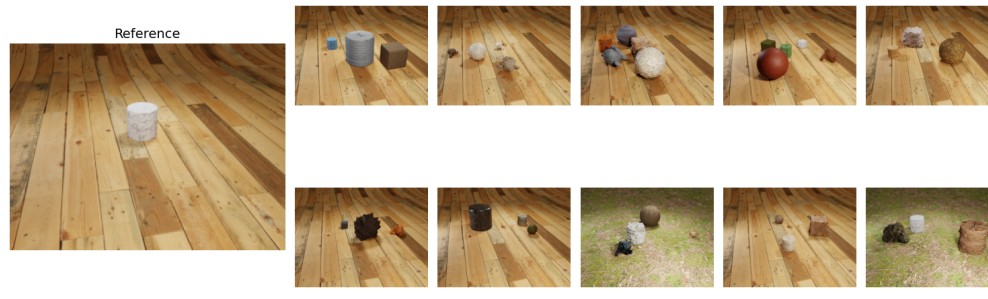

Top-10 Retrievals with Oh-A-DINO

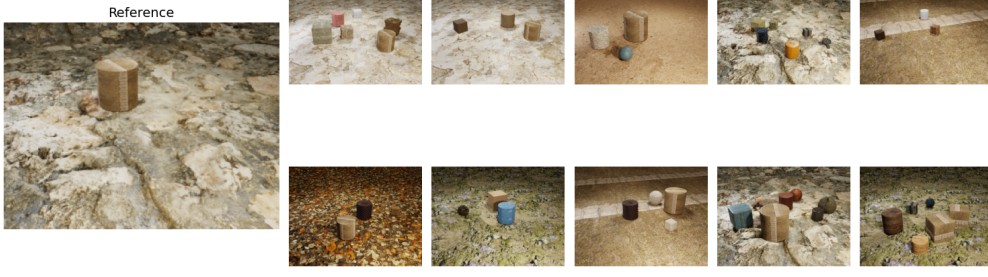

Top-10 Retrievals with DINOv2

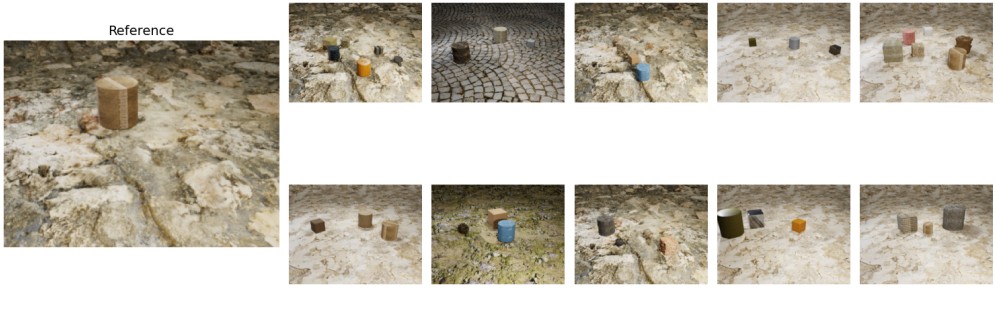

Top-10 Retrievals with VAE

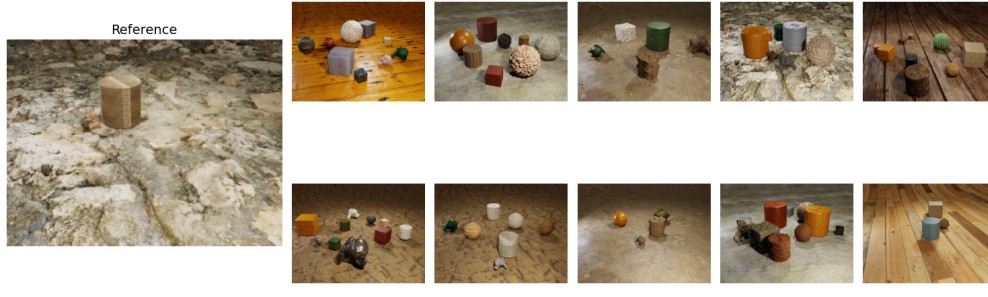

Top-10 Retrievals with SlotDiffusion

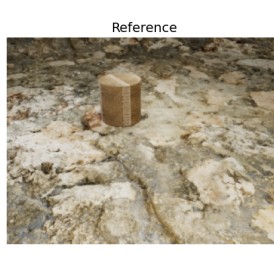
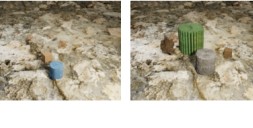
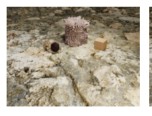
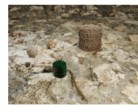
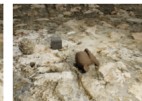
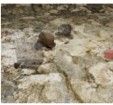
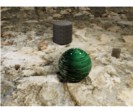
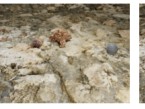
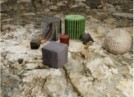
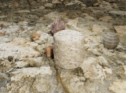

Reference

## E.4 CLEVR

Compared to CLEVRTex the differences in retrieval performance are more nuanced. We see that Oh-A-DINO is able to retrieve all objects almost without errors, compared to the DINOv2 representations, which again retrieves the correct objects, but with wrong color or sometimes object properties. The VAE object-level features perform well in CLEVR due to the lack of distractors such as the background, however when comparing to the Oh-A-DINO retrievals, we observe that the retrievals with VAE respect the scene structure less. For instance, at times the object in the retrieval is not in the same position, orientation as the reference object or the same object is present multiple times in the scene. Again, SlotDiffusion performs the worst. Compared to CLEVRTex we can see that the SlotDiffusion representations retrieve similar objects, but often fails to bind different attributes together resulting in difficult to interpret retrievals, confirming our numerical results.

Top-10 Retrievals with Oh-A-DINO

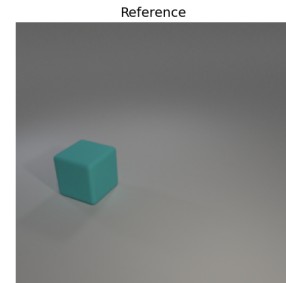
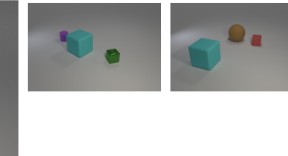
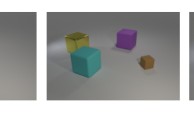
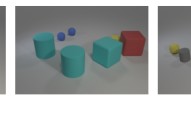
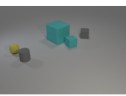
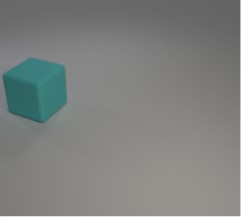
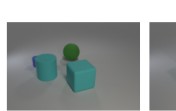
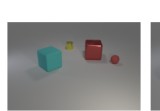
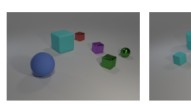
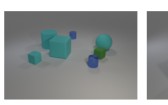
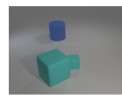

Reference

Top-10 Retrievals with DINOv2

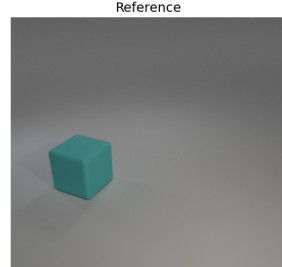
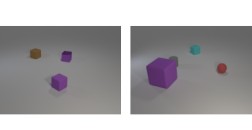
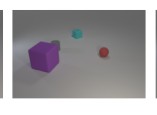
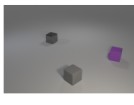
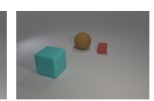
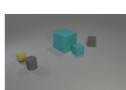
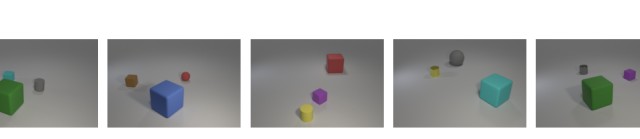

Reference

Top-10 Retrievals with VAE

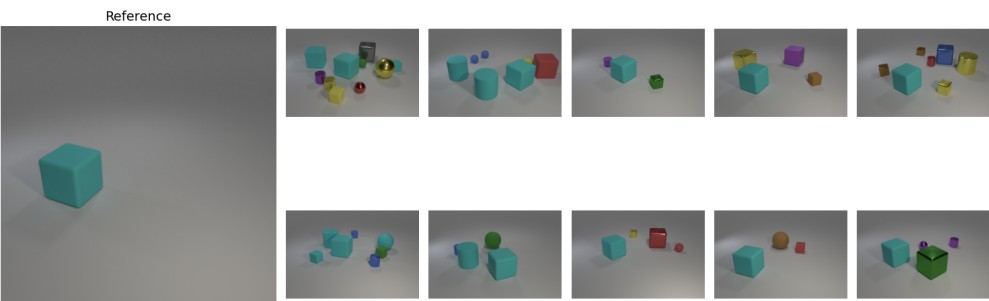

Top-10 Retrievals with SlotDiffusion

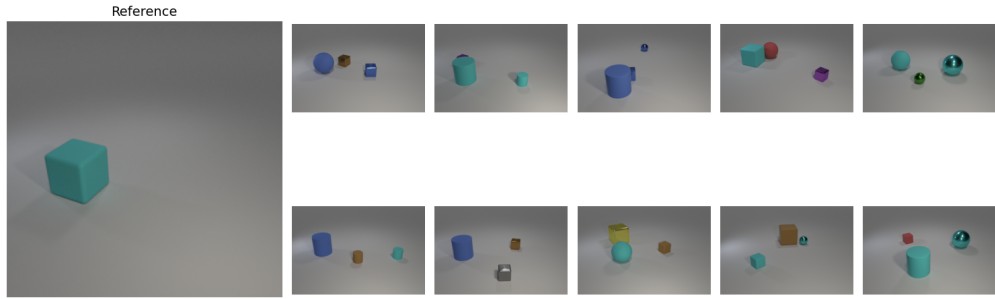

Top-10 Retrievals with Oh-A-DINO

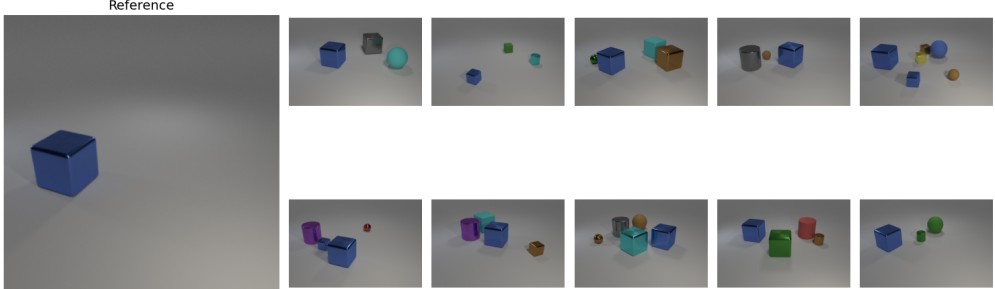

Top-10 Retrievals with DINOv2

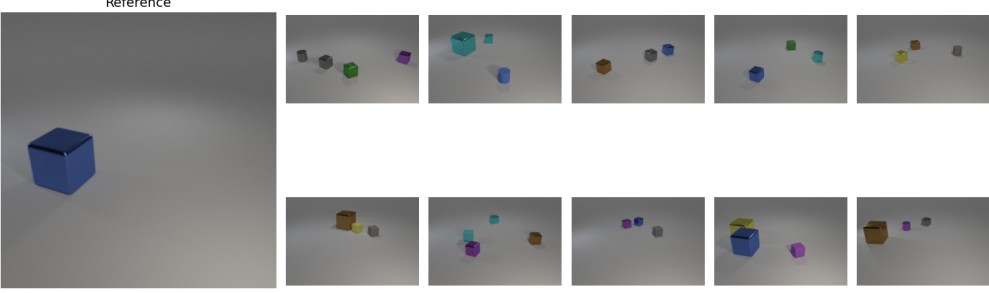

Top-10 Retrievals with VAE

Reference

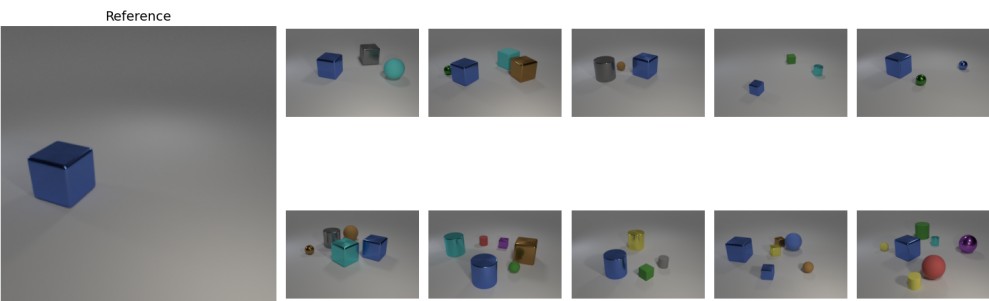

Top-10 Retrievals with SlotDiffusion

Reference

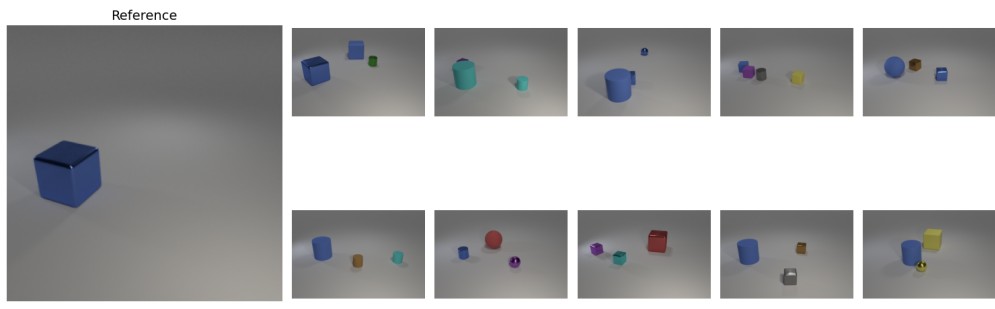

Top-10 Retrievals with Oh-A-DINO

Reference

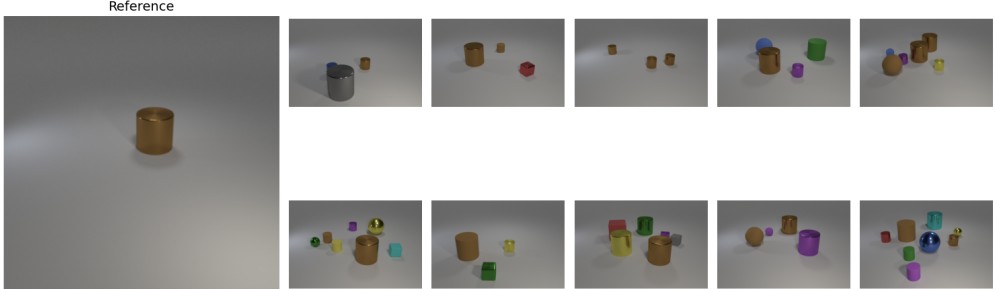

Top-10 Retrievals with DINOv2

Reference

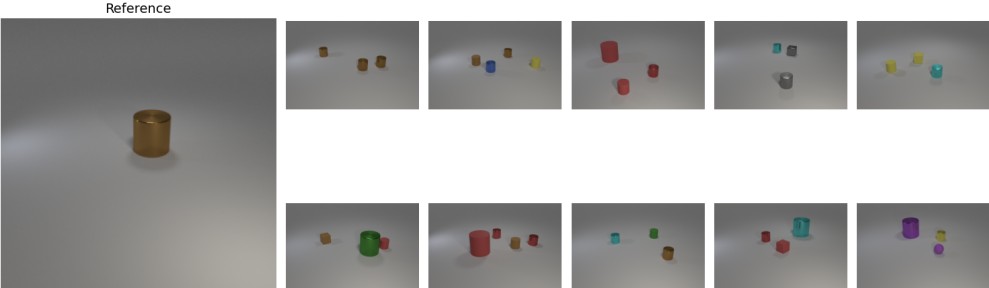

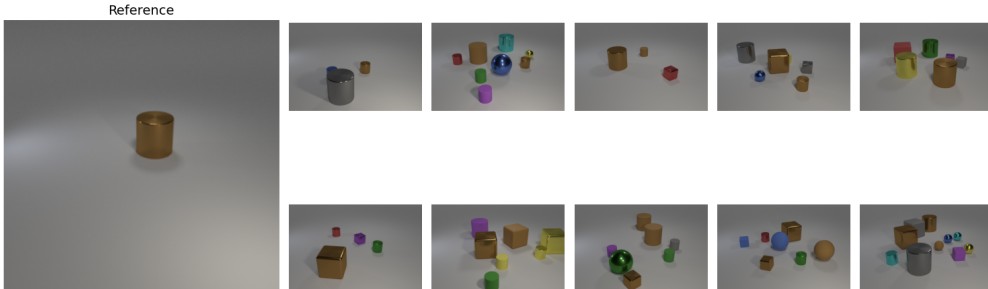

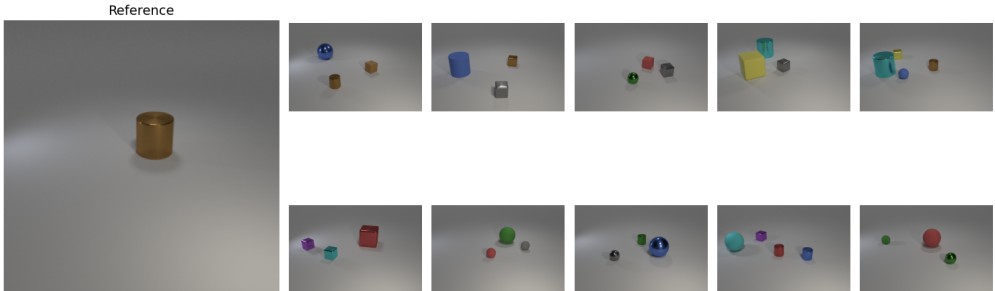

