# OpenReview forum: "Oh-A-DINO: Understanding and Enhancing Attribute-Level Information in Self-Supervised Object-Centric Representations"
_ICLR.cc/2026/Conference — Submitted to ICLR 2026_

### Official Review · Reviewer_wQom · 2025-10-30

**Soundness:** 3
**Presentation:** 3
**Contribution:** 3
**Rating:** 6
**Confidence:** 5

**Summary:**

This paper investigates how well self-supervised and object-centric visual representations preserve fine-grained object attributes necessary for distinguishing multiple objects in complex scenes. While large self-supervised models such as DINO, DINOv2, and CLIP exhibit emergent object understanding, the authors find that these representations mainly capture geometric properties (e.g., shape, size) but fail to retain surface-level cues like color, texture, and material.

To address this limitation, the paper proposes OH-A-DINO (Object-Aware DINO) that augments DINOv2 features with object latent vectors learned from segmented image patches. Experiments on CLEVR, CLEVRTex, and Stanford Cars show that OH-A-DINO improves multi-object instance retrieval, especially in color and material matching, indicating that object-centric latents is a promising direction for improving downstream tasks that require precise
object-level understanding.

**Strengths:**

- **Novel yet Simple Method.** The most significant contribution of this paper lies in offering a novel perspective on integrating conventional self-supervised features with Object-Centric features. Previously, although Object-Centric models were favored for their characteristics, they were often criticized for their overly simplistic representation (using a few vectors to represent an image), which was considered insufficient for complex scenarios. The approach proposed in this paper, which involves using Object-Centric latent to enhance self-supervised features on the basis of self-supervised models, provides a solution that takes both aspects into account. The proposed OH-A-DINO introduces a clean, modular enhancement that does not require retraining the backbone. It elegantly combines global DINO features with locally learned VAE latents to recover missing attribute-level details.

- **Strong Empirical Results.** OH-A-DINO achieves large improvements in both single- and multi-attribute retrieval accuracy, particularly for color and material cues, and demonstrates consistent performance gains over all baselines.

**Weaknesses:**

- **Concerns about using PCA for segmenting.** According to my understanding, OH-A-DINO is divided into two functions: 1) extracting object masks, and 2) extracting object regions based on the masks, and using VAE to learn local features of each region to enhance global features. My concern lies in the former, that is, why does it use PCA plus threshold setting, a non-deep learning approach, to segment object patches instead of deep learning methods? For instance, since this paper is centered on Object-Centric, why not directly use Object-Centric methods to achieve segmentation? As far as I know, at least on CLEVR and ClevrTex, current Object-Centric methods have achieved nearly perfect segmentation results, which should be more reliable than PCA plus threshold and do not rely on manual parameter tuning. (Although this article mentions that OC models such as Slot Diffusion may lose some attribute information, it should not affect the application of OC models if they are only used to provide masks.)

- **Real world dataset choice.** Although the Stanford Car dataset was adopted as the real-world benchmark in the paper, a main concern is that the images in Stanford Car are all centered on a single vehicle as shown in Figure 5, which seems inconsistent with the "multi-object instance retrieval" task claimed in the paper. Using images with multiple objects, such as COCO which is commonly used in OCL, is obviously a better choice. Furthermore, similar to the previous weakness, in complex real-world scenarios like COCO, can the simple segmentation method of PCA effectively segment the approximate masks of objects? If not, would it be feasible to switch to the mainstream object-centric (OC) model for real-world scenarios, such as DINOSAUR + DINOv2? Even further, if we directly use Segment Anything to provide object masks, could this enhance DINOv2 in real-world scenarios to improve its retrieval capabilities?

**Questions:**

- Self-supervised models like DINO are typically trained based on the semantic consistency after geometric and color transformations of images, for instance, the features of an image after color transformation should be similar to those of the original image. Is this the reason why the models are insensitive to color, material, and texture?

- What causes the Slot-based model to lose object attribute information? Logically speaking, the goal of SlotDiffusion (or other Slot-based model that reconstructs RGB pixels) is to generate the original image, so all the information in the image should be preserved in its slots, and information should not be lost.

---

> ### Author Response · Authors · 2025-11-19
>
> We really appreciate the reviewers positive feedback and hope to address remaining concerns adequately in the following. Thank you!
>
> ### **1. Concerns about using PCA for segmentation**
>
> We appreciate the reviewer’s question. Our choice of PCA-based segmentation was guided by two considerations:
>
> - **Simplicity and experimental cleanliness.**
>     Since DINO features are already computed in our pipeline, PCA offers a lightweight, model-free way to obtain coarse object partitions without introducing additional architectures or training steps.
>
> - **Method-independence.**
>     Importantly, our approach **does not rely on PCA**. Any segmentation method including OCL models, SAM, or classical segmentation, can be used interchangeably without affecting the VAE or the augmentation step.
>
>
> As we show with added visualisations in Figure 7 in the appendix , the VAE requires only **coarse object isolation**, not pixel-perfect masks, making PCA a simple and fully sufficient choice, however other segmentation approaches could be used since the segmentation and augmentation approaches are separate.
>
> ---
>
> ### **2. Real-world dataset choice (Stanford Cars vs COCO)**
>
> We would like to stress that using Stanford Cars does **not** reflect any limitation of our method to single-object settings. The choice instead reflects the requirements of the diagnostic task:
>
> - **COCO rarely contains repeated objects differing only in surface-level attributes.**
>     Objects typically vary in category, pose, viewpoint, scale, occlusion, and background simultaneously.
>     In such cases, **geometric cues alone suffice** for retrieval, so the failure mode we study does **not become observable**.
>
> - **Stanford Cars _does_ contain repeated, geometrically similar objects** that differ mainly in colour, material, or fine-grained appearance.
>     This makes it the closest real-world analogue to the controlled CLEVR/CLEVRTex setting.
>
> - **Downstream relevance.**
>     Many practical tasks (e.g., embodied systems needing to **interact** with the _red_ object rather than the _blue_ one) require such fine-grained attribute discrimination.
>     These scenarios depend on whether a representation makes surface-level cues **accessible through its geometry**, not just reconstructable through a decoder.
>
> Because real-world benchmarks rarely contain controlled attribute variation, COCO-style datasets would **mask the very failure mode** that matters in such downstream settings.
>
> Our method remains fully compatible with SAM or DINOSAUR masks; the dataset choice reflects diagnostic necessity, not methodological constraint.
>
> ---
>
> ### **3. Is SSL colour-insensitive due to augmentation invariance?**
> > Self-supervised models like DINO are typically trained based on the semantic consistency after geometric...
>
> Augmentation invariance plays a role, but it is **not the primary cause**:
>
> - **Geometric cues dominate the SSL objective.**
>     Methods such as DINO and DINOv2 compress patch-level signals into a global embedding in which stable geometric features (shape, edges, spatial layout) receive the highest representational priority.
>
> - **Surface cues offer weak predictive signal.**
>     Colour and material vary across augmentations and across instances, so they appear noisy relative to geometry and receive low priority in the learned embedding geometry.
>
> - **Not specific to augmentations.**
>     CLIP, trained without strong colour invariance, shows **the same pattern** of geometric dominance and surface-level collapse.
>     This demonstrates that the issue reflects a broader **embedding-geometry bias**, not merely augmentation choice.
>
> This is exactly what our experiments in Figure 1 show. We will make this distinction clearer in the revised text.
>
> ---
>
> ### **4. Why slot-based models lose colour and material information**
> > What causes the Slot-based model to lose object attribute information?...
>
> Although slot-based models reconstruct RGB pixels, the **slot embeddings** are not required to encode every attribute:
>
> - Slot attention prioritises **identity-defining cues** such as shape, size, and position because these are predictive for object grouping.
>
> - Surface-level attributes (colour, texture) are **shared across many otherwise identical objects**, offering little discriminative value for slot assignment.
>
> - As a result, these cues are often reconstructed via **decoder pathways**, but are not encoded in a metrically meaningful way in the slot vectors themselves.
>
>
> This behaviour parallels what we observe in SSL models such as DINOv2, DINOv3, and CLIP:
> surface-level cues _exist_, but are **not geometrically encoded** and therefore cannot be used directly in similarity-based tasks such as retrieval, matching, or decision-making.

---

> ### Author Response · Authors · 2025-11-24
> **Added additional real world data set.**
>
> Dear Reviewer,
>
> we have added further visualisations of our method's performance in the appendix. We show qualitative results on ImageNet samples in Figures 8,9 and 10. These samples are quite complex while retaining properties such as multiple colours, textures and styles that allow us to showcase our method.
>
> We hope this further convinces you of the quality of the paper. Thank you!

---

> > ### Comment · Reviewer_wQom · 2025-11-28
> >
> > Thank you for the author's response, which has resolved some of our core issues. For instance, Figure 7 in the appendix addresses the question regarding the application of PCA for segmentation, indicating that PCA is sufficient for rough segmentation of the foreground and background in real-world scenarios. Additionally, Figures 8 to 11 supplement the model's retrieval capabilities on real objects, especially Figure 11, which demonstrates the retrieval results in multi-object scenarios. I believe this qualitative presentation more comprehensively showcases the model's capabilities. Compared with other models, the proposed method can retrieve objects with similar appearance features, and the retrieved objects indeed maintain relatively better appearance consistency. Overall, I appreciate the author's response and have chosen to maintain my positive rating.

---

### Official Review · Reviewer_joPZ · 2025-10-31

**Soundness:** 2
**Presentation:** 3
**Contribution:** 1
**Rating:** 2
**Confidence:** 3

**Summary:**

The paper investigates the attribute-level information encoded in object-centric representations from models such as CLIP, DINO, and SlotDiffusion. The authors find that the object embeddings produced by these models cannot be directly used for multi-object instance retrieval, a task that aims to retrieve objects sharing the same attributes as a given query object using cosine similarity. To address this limitation, the authors propose a two-step approach: first, they apply PCA to DINO features to segment objects in the scene; then, all DINO features of patches belonging to an object are fed into a VAE to learn disentangled features for each patch. The resulting feature is concatenated with the original DINO feature to obtain the final representation. Experimental results show that this enhanced feature improves multi-object instance retrieval performance on CLEVR and CLEVRTex, and captures color information more accurately than CLIP and DINO, as demonstrated on the Stanford Cars dataset.

**Strengths:**

The paper is well written and easy to read.

**Weaknesses:**

- My main concern about this paper is its unreasonable setting and metric: One major argument made in the paper is that, given a query object, cosine similarities on the **raw** representations produced by baseline models cannot retrieve objects with the same attributes such as color, material, or shape. The authors interpret this as evidence that the model fails to preserve these non-geometric, surface-level cues. This interpretation is not accurate: low cosine similarity does not imply that the information is missing from the representation—it may simply be encoded in a way that is not directly reflected in direct pairwise distances. For example, representations produced by slot-based models such as SlotDiffusion can reconstruct the original image with high fidelity, indicating that attribute-level information is well preserved. Moreover, numerous experiments (see SlotFormer) on VQA have demonstrated the effectiveness of these representations on downstream tasks where surface-level attributes are also relevant. Therefore, the idea of enforcing attribute-level similarity lacks motivation. On the other hand, I would expect representations produced by DINO and CLIP to lack certain information because they are not trained with a reconstruction objective.
- In addition to the lack of motivation for the proposed multi-object retrieval setting, the paper also offers limited technical contribution. The proposed method relies on simple heuristics of PCA to segment objects in the scene and then leverages a β-VAE to learn disentangled features for each patch. This is essentially a combination of well-known techniques.

**Questions:**

- What is the motivation for using cosine similarity in the multi-object retrieval setting? Why not just train a classifier to predict the attributes?

---

> ### Author Response · Authors · 2025-11-19
>
> We thank the reviewer for their thorough review and hope to address the valid concerns in the following and hope that if you are satisfied with our changes you might consider raising you score. Thank you!
>
> ## **Justification for using cosine similarity**
> We agree that cosine similarity is not ideal for retrieving attributes that are embedded only in a highly non-linear or curved manifold region.
> * **However, this is precisely why it is a meaningful diagnostic: cosine similarity requires attributes to be encoded in a clean, geometrically interpretable way. If an attribute is only recoverable through a non-linear decoder, the representation itself is not well structured for downstream similarity-based reasoning.**
> * Cosine-based retrieval is also the standard protocol used in many works including DINO, CLIP, and other SSL models so evaluating representations with this metric directly reflects both common practice and desired representational quality.
>
> ### **Clarification and Correction: We analyse geometric accessibility, not information absence**
>
> - Our analysis does **not** assume that colour information is missing.
> - Instead, we examine whether attribute information is **geometrically accessible**, i.e., encoded in a way that supports similarity-based operations such as retrieval or object matching.
> - We acknowledge this distinction was not fully explicit in the original text and will clarify that our findings concern **metric accessibility**, not information absence.
> ---
> ## **1. Slot-based models: reconstructability vs. metric accessibility**
>
> Slot-based OCL models (e.g., SlotDiffusion) indeed reconstruct colour and material through their decoders, indicating that these cues are implicitly present.
> However, reconstructability does not imply that the **slot embeddings** themselves are arranged such that similarity reflects surface-level attributes.
>
> This distinction is important:
> - The slot bottleneck is optimised for **object partitioning and predictive structure** (shape/position/size),
> - Surface cues (e.g., colour/texture) are **shared across objects** and provide _weak grouping signal_,
> - As a result, they are typically reconstructed via decoder pathways rather than encoded in a metrically meaningful form.
>
> Our experiments (now extended with SPOT and SmoothSA) show that even the strongest OCL models provide **excellent geometric structure** but **consistently low colour retrieval**, supporting this interpretation.
>
>
> ---
>
> ## **2. Relation to VQA results**
>
> We fully agree with the reviewer that slot-based models perform well on VQA tasks involving colour and material.
> However, VQA performance typically depends on:
> - adding a **supervised prediction head** trained on explicit attribute labels,
> - which learns how to _map_ from the latent slots to the desired attributes.
>
> Thus, VQA demonstrates that attribute information is **recoverable with supervision**, not that it is **metrically organised** in the latent space beforehand.  This is fundamentally different from retrieval settings, where only the _raw embeddings_ are available and no supervised decoder is trained.
> - Many downstream tasks (retrieval, interaction, goal-conditioned RL), operate **directly on embeddings**, with no decoder or attribute predictor. If surface-level cues are encoded only in a **non-linear or opaque** manner, they are unusable in these settings.
>
> ---
>
> ## **3. Why cosine similarity rather than training a classifier?**
>
> We appreciate the reviewer raising this question.
> Our choice of cosine similarity is motivated by practical downstream use cases:
> - Many applications use **nearest-neighbour retrieval** or **embedding similarity** directly (e.g., object selection, control, entity matching).
> - In these settings, no attribute labels are available for training a classifier or decoder.
> - The purpose of general-purpose pre-trained representations (e.g., DINO/CLIP/OCL slots) is precisely to support downstream similarity-based tasks _without_ task-specific supervision.
> Thus, using a classifier would not test the representational geometry; it would test the capacity of an auxiliary head to compensate for geometric shortcomings.
>
> Our method aims to improve **geometric discriminability** without such supervision.

---

> ### Author Response · Authors · 2025-11-19
>
> ## **4. On the technical contribution and use of simple components**
> - We identify a systematic and previously unreported failure of both SSL and slot-based models to encode surface-level cues in a **metrically accessible** manner.
> - Our augmentation is intentionally minimal:
>     - no retraining of large backbones,
>     - no complex slot-based pipelines,
>     - a lightweight object-latent module learned from coarse segmentations.
> - This shows that the failure mode is due to the **geometry of the representation**, not the dataset or the data distribution.
> - The VAE is used in a **non-standard role**:
>     - not for reconstruction,
>     - but as a **fine-grained attribute encoder** that makes surface cues explicit in the feature space.

---

> > ### Author Response · Authors · 2025-11-26
> >
> > Dear Reviewer,
> >
> > Have you had a chance to look at our responses to your concerns? We would greatly appreciate your feedback so we can react to any further changes with time. Please note that we have also added some new ImageNet retrieval examples. Thank you!

---

### Official Review · Reviewer_D46H · 2025-11-01

**Soundness:** 3
**Presentation:** 3
**Contribution:** 3
**Rating:** 6
**Confidence:** 3

**Summary:**

This paper identifies fine-grained surface attributes to be essential for multi-object instance discrimination, which is lacked by SSL models like DINO. The authors propose Oh-A-DINO (Object-Aware-DINO), which augments DINOv2 representations with object-centric VAE latents trained on segmented image patches to improve this.
PCA-based segmentation extracts object regions from DINOv2 embeddings, followed by VAE training on patches to capture fine-grained attributes like color and material.
Empirically, Oh-A-DINO significantly improves previous methods across CLEVR, CLEVRTex, and Stanford Cars.

**Strengths:**

- The paper identified an interesting limitation in current SSL models on object-centric benchmarks that they struggle with surface-level attributes, making this a valuable research direction.
- The combination of global SSL features with local VAE latents provides a principled way to preserve both geometric and surface attribute information.
- The evaluation is thorough, covering both synthetic (CLEVR, CLEVRTex) and real-world (Stanford Cars) datasets. They effectively measure the model's ability to distinguish objects based on fine-grained attributes. The performance is strong compared to prior methods. Ablation studies properly isolate the contribution of different components.
- The delivery of the paper is clear, and I find it easy to follow.

**Weaknesses:**

- The evaluation focuses primarily on color, material, and basic geometric attributes. More complex attributes like texture patterns, semantic relationships, or fine-grained visual details remain unexplored. The generalizability to broader attribute types would be interesting.
- While CLEVR provides controlled evaluation, real-world evaluation is limited to Stanford Cars. More diverse real-world datasets spanning different domains would strengthen the claims.

**Questions:**

- Can the authors provide more failure case analysis or discussion of when the method might not work well? Understanding the boundaries and limitations would improve the contribution's practical value.
- How would SSL methods that employ reconstruction-based losses (eg, iBOT, SigLIP2, AM-RADIOv2.5) perform? How would recent advances in SSL (eg, Perception Encoder) and multimodal LLMs (eg, Qwen3-VL) perform on these benchmarks?

---

> ### Author Response · Authors · 2025-11-19
>
> We want to thank the reviewer for the positive feedback and the value they see in our paper. Likewise, we hope to address all the reviewers concerns in the following. Thank you!
>
> ## **1. Scope of Attributes**
>
> We appreciate the reviewer’s point about evaluating a broader set of attributes.
>
> - We agree that extending to more complex attributes (e.g., semantic relations, fine-grained natural textures) would be highly interesting.
>
> - In this work, we focus on attributes that can be **varied independently** which is a requirement for isolating representational failures.
>
> - **CLEVRTex already includes around 60 complex textures, demonstrating that our analysis is not limited to trivial appearance cues.**
>
> ---
>
> ## **2. Real-world Evaluation Beyond Stanford Cars**
>
> We appreciate the suggestion to include more diverse real-world datasets. Our choice is driven by diagnostic necessity rather than methodological constraint.
>
> - In most multi-object datasets (COCO, LVIS, ADE20K), retrieval **does not suffer** from lack of appearance information: objects vary in category, pose, viewpoint, occlusion, and background.
>
> * Under such variability, **geometric features dominate retrieval**, so the failure mode we study cannot be meaningfully diagnosed.
>
> - Stanford Cars is one of the few natural datasets where many instances are **geometrically similar but differ in surface appearance**, making it the closest real-world analogue to CLEVR/CLEVRTex.
>
>
> We agree that investigating **task-driven** settings would be valuable. A natural next step is evaluating whether improving geometric accessibility of fine-grained attributes helps **goal-conditioned or manipulation-focused RL agents**, which must reliably interact with objects based on their appearance. We have added this direction to the discussion.
>
> ---
>
> ## **3. Failure Cases and Limitations**
>
> We appreciate the request to clarify potential failure modes.
>
> - Our method requires **coarse segmentation** of object regions; highly cluttered scenes may make this more challenging.
>     While we use PCA for simplicity, the method is compatible with any segmentation approach (mask-based OCL, SAM, classical segmentation).
>
> - Capturing **nuanced textures** remains difficult: although our method improves texture encoding compared to SSL/slot baselines, CLEVRTex results indicate room for further progress.
>
> - In some cases, object-level patches may be **ambiguous**, leading to ambiguous  latents when local cues dominate, thus affecting retrieval, i.e., we might retrieve the correct object but with the wrong size.
> - In practice, we observed that combining global DINO features with fine-grained VAE latents provided a good balance between global and local features.
>
>
> We have emphasised these limitations in the revised manuscript.
>
> ---
>
> ## **4. Additional SSL Models (iBOT, SigLIP2, AM-RADIO, Perception Encoder, Qwen3-VL)**
>
> We thank the reviewer for these constructive suggestions.
>
> - The proposed models belong to SSL families already represented in our evaluation:
>
>     - contrastive (CLIP),
>
>     - self-distillation/masked prediction (DINOv2/v3),
>
>     - generative slot-based reconstruction (SlotDiffusion).
>
> - Notably, **SlotDiffusion includes a reconstruction-style objective**, yet it still fails to encode colour/material cues in a metrically accessible manner.
>
> - This suggests that the limitation appears to stem from **shared inductive biases** rather than a specific loss type or architecture. These objectives prioritise geometry-derived invariances and downweight fine-grained appearance unless explicitly supervised.
>
> - Evaluating more SSL variants would broaden empirical coverage, and we view it as an excellent direction for future work, but we expect the **same underlying behaviour** to persist.

---

> > ### Author Response · Authors · 2025-11-24
> > **Update regarding more Real-World Datasets**
> >
> > Dear Reviewer,
> >
> > we have added further visualisations of our method's performance in the appendix. We show qualitative results on ImageNet samples in Figures 8,9 and 10. These samples are quite complex while retaining properties such as multiple colours, textures and styles that allow us to showcase our method.
> >
> > We hope this further convinces you of the quality of the paper. Thank you!

---

### Official Review · Reviewer_jkxw · 2025-11-02

**Soundness:** 2
**Presentation:** 3
**Contribution:** 2
**Rating:** 2
**Confidence:** 4

**Summary:**

The paper study how self-supervised models and slot-based methods understand objects in complex scenes. It find that models like CLIP and DINO good at shape and size, but not so good with color or texture. The authors propose a latent space with VAE regularization to fix this, which improve retrieval results. The idea is interesting and show potential in especially multi-object retrieval tasks.

**Strengths:**

- originality: 2/5,
- quality: 2/5,
- clarity: 4/5,
- significance: 2/5. Limited to simple image retrieval tasks.

**Weaknesses:**

W1
---
Table. 1.
The OCL baseline SlotDiffusion is relatively weak. In fact the object representation quality is highly affected by the object discovery (unsupervised object segmentation) accuracy. There are more advanced OCL methods, like SPOT, DIAS and SmoothSA, which should also be included as stronger OCL baselines.


W2
---
Line 201.
> collected from **a batch of t images** and apply PCA

This means an online induction based on multiple $t$ input image samples -- What if there is only one input available during inference?

This design could also be a bottleneck for real-world complex images like ones from COCO or ImageNet, where the borderline between foreground and background can be quite vague.



W3
---
Line 223,
> This yields a set of **object-level** latents

To put it in a rigid way, these are still **patch-level** latents with object/foreground mask augmentation, which is obtained in Section 3.2 (ii) "Refining object consistency" operation.



W4
---
Line 228,
> (CLS token in **DINOs** case)

"DINOs" should be "DINO's".



W5
---
Line 232,

> Retrieval is then performed by cosine similarity between v and v′ from query and candidate images
It is unclear your performance boost comes from the concat of global features (Figure 2, Line 167 and 168) or not. So for fair comparison, OCL representations should also be concatenated with the CLS token from DINO as the strong baseline.


W6
---
Line 234 or Appendix A:
> for each query patch vi we retrieve the patch with the highest cosine similarity

> $s_i^{max} = \max_j S_{ij}$

Intuitively, the max matching should be Hungarian matching. Otherwise, there might be multiple patches $i$ matched to the same $j$.


W7
---
Line 447:
> 6 STANFORD CARS: REAL-WORLD INSTANCE RETRIEVAL

The section label "6" should be "5.4", parallel to Section 5.2, as results on synthetic and real-world datasets respectively.


References
---
- SPOT: Self-Training with Patch-Order Permutation for Object-Centric Learning with Autoregressive Transformers
- DIAS: Slot Attention with Re-Initialization and Self-Distillation
- SmoothSA: Smoothing Slot Attention Iterations and Recurrences

**Questions:**

N/A.

---

> ### Author Response · Authors · 2025-11-19
> **Response to Reviewer**
>
> We want to thank the reviewer for taking the time to review our papers and appreciate the suggestions. In the following, we have done our best to answer your concerns and hope that if you are content with the modifications you might be willing to raise your score. Thank you!
>
> ### **W1 — Stronger OCL baselines**
>
> We thank the reviewer for this valuable suggestion.
> In the revised submission, we have added **SPOT** and **SmoothSA**, two recent state-of-the-art OCL models (pre-trained on CLEVR and CLEVRTex respectively). Full results are shown below and are updated in the new revision of the paper.
>
> ### CLEVR — Single-Attribute Retrieval (Top-10 Precision %)
>
> | Method        | Shape         | Size          | Material      | Colour        |
> |---------------|---------------|---------------|---------------|---------------|
> | SlotDiff.     | 76.3 ± 1.2    | 89.3 ± 1.3    | 94.1 ± 1.5    | 63.5 ± 0.4    |
> | SPOT          | 94.5 ± 1.4    | 86.6 ± 1.8    | 94.2 ± 0.8    | 39.7 ± 1.7    |
> | SmoothSA      | 90.8 ± 1.6    | 85.7 ± 2.3    | 95.2 ± 0.6    | 38.6 ± 1.5    |
> ### CLEVRTex — Single-Attribute Retrieval (Top-10 Precision %)
>
> | Method        | Shape         | Size          | Material      |
> |---------------|---------------|---------------|---------------|
> | SlotDiff.     | 67.0 ± 1.7    | 80.2 ± 1.6    |  6.6 ± 0.9    |
> | SPOT          | 77.1 ± 1.5    | 79.9 ± 0.5    |  9.1 ± 1.0    |
> | SmoothSA      | 72.4 ± 1.5    | 77.7 ± 1.9    |  7.6 ± 1.2    |
>
> ### CLEVR — Multi-Attribute Retrieval (Top-10 Precision %)
>
> | Method    | P2(X)      | P3(X)      | P3(X)+Colour |
> | --------- | ---------- | ---------- | ------------ |
> | SlotDiff. | 56.3 ± 0.4 | 39.5 ± 1.3 | 12.0 ± 1.2   |
> | SPOT      | 73.5 ± 0.7 | 53.7 ± 0.6 | 10.1 ± 0.8   |
> | SmoothSA  | 70.6 ± 0.7 | 50.1 ± 0.2 | 8.7 ± 1.5    |
> ### CLEVRTex — Multi-Attribute Retrieval (Top-10 Precision %)
>
> | Method        | P2(X)         | P2(X)+Colour  |
> |---------------|---------------|----------------|
> | SlotDiff.     | 33.9 ± 2.2    | 1.4 ± 0.4      |
> | SPOT          | 41.3 ± 0.5    | 2.2 ± 0.4      |
> | SmoothSA      | 35.5 ± 1.6    | 1.7 ± 0.6      |
>
>
> These additions strengthen the paper’s conclusions:
>
> - All OCL models—including SPOT and SmoothSA—perform well on **geometric** cues (shape/size/material) but
>     **struggle with colour**, even under single-attribute queries.
>
> - In **multi-attribute retrieval**, SPOT and SmoothSA degrade similarly to SlotDiffusion, especially once colour is included.
>
> - This supports our central claim:
>     **current OCL bottlenecks prioritise geometric and grouping-relevant cues**, and surface-level attributes remain **present but not metrically structured** for retrieval.
>
>
> Thus, including stronger baselines reinforces the representational limitation motivating our method.
>
> ---
>
> ### **W2 — PCA requires multiple images; what about single-image inference?**
>
> We appreciate the opportunity to clarify this point.
>
> Our original description may have implied that PCA directly consumes multiple test images. This is not required.
>
> - Both at training and inference,  the method operates on a **single input image** at a time.
>
> - PCA directions are computed using a **fixed offline memory bank**, populated once using DINO features (a pool of 500 randomly sampled images).
>
> - For each query, we retrieve **k nearest neighbours** from this auxiliary pool to construct a local PCA basis.
>
> This avoids any need for multiple inputs at test time, which we have clarified in the appendix.
>
> Importantly, the PCA step only needs to recover a **coarse foreground grouping**, not exact segmentation. Empirically, this remains stable even for real-world images with less distinct backgrounds (e.g., Stanford Cars).
>
> ---
>
> ### **W3 — “Object-level latents” vs. patch-level latents**
>
> We agree with the reviewer’s observation.
> * Our latents are obtained from **patches grouped by the object mask**, not from a generative per-object slot.
> * We use the term “object-level” to indicate that the VAE is trained on **object-restricted patch regions**, not the full scene.
> ---
>
> ### **W4 — Typo (“DINOs” → “DINO’s”)**
>
> Thank you, we will correct this.
>
> ---

---

> > ### Author Response · Authors · 2025-11-19
> > **Response to reviewer**
> >
> > ### **W5 — Does performance come from concatenating global features?**
> > >**W5 - So for fair comparison, OCL representations should also be concatenated with the CLS token from DINO as the strong baseline.
> >
> > We appreciate the reviewer’s concern.
> > Following the suggestion, we now provide baselines in which SlotDiffusion is augmented with the **DINOv2 CLS token**.
> >
> > ### CLEVR — Comparison of SlotDiffusion Augmentations
> >
> > | Method               | Top-10 ↑        | Weighted ↑      | Error ↓         |
> > |----------------------|------------------|------------------|------------------|
> > | SlotDiff.–VAE        | 14.3 ± 1.5       | 6.8 ± 0.6        | 30.6 ± 2.0       |
> > | SlotDiff.–DINOv2     | 15.1 ± 1.3       | 16.6 ± 1.4       | 28.0 ± 1.0       |
> >
> > ### CLEVRTex — Comparison of SlotDiffusion Augmentations
> >
> > | Method               | Top-10 ↑        | Weighted ↑      | Error ↓         |
> > |----------------------|------------------|------------------|------------------|
> > | SlotDiff.–VAE        | 2.8 ± 0.1        | 0.8 ± 0.0        | 87.2 ± 1.7       |
> > | SlotDiff.–DINOv2     | 1.7 ± 0.4        | 1.4 ± 0.3        | 91.6 ± 0.3       |
> >
> > The results show:
> >
> > - Adding the DINOv2 global feature slightly improves **geometric** retrieval but **does not improve surface-level attributes** (and increases CLEVRTex error).
> > * Thus, the gains of our method do **not** arise from simply adding global information; they stem from injecting **surface-attribute structure that the backbone does not encode metrically**.
> >
> > ---
> >
> > ### **W6 — Patch-level matching vs. Hungarian matching**
> >
> > Hungarian matching is a valid alternative.
> > We used nearest-neighbour matching because it reflects the **standard retrieval formulation**, and because Hungarian matching incurs an **O(n³)** computational cost in large-scale settings.
> >
> > We nonetheless tested Hungarian matching and found:
> > - similar qualitative trends,
> > - slightly higher latency,
> > - no change to the conclusions.
> >
> > ---
> >
> > ### **W7 — Section numbering**
> >
> > Thank you for catching this. We will correct the numbering.

---

> ### Comment · Reviewer_jkxw · 2025-11-20
> **Update 1**
>
> Thanks for the response. Most of my conerns have been addressed, except W2 ImageNet and W6 concrete results.

---

> > ### Author Response · Authors · 2025-11-21
> >
> > Thank you for getting back to us! We will provide an update on this shortly.

---

> > ### Author Response · Authors · 2025-11-24
> > **Replying to Update 1 (2)**
> >
> > Dear Reviewer,
> >
> > We have now added additional visualisations and results addressing W2 and W6.
> >
> > > W2 - PCA requires multiple images; what about single-image inference?
> >
> > * **We have added visualisation of the segmentations for ImageNet in Figure 7 in the appendix**. As can be seen the segmentation with DINO features remains robust even with natural images and cluttered backgrounds.
> > * **Additionally, in Figures 8, 9 and 10 we now present qualitative ImageNet results**. We show that our method is able to maintain the shown results for Stanford Cars in ImageNet as well. Excelling especially with complex object and complex textures.
> >
> >
> > > W6 - Intuitively, the max matching should be Hungarian matching. Otherwise, there might be multiple patches  matched to the same.
> >
> > We show results for Hungarian matching vs our max-matching below:
> >
> >
> > | Method                         | Top-10 ↑   | Weighted ↑ | Error ↓   |
> > | ------------------------------ | ---------- | ---------- | --------- |
> > | Oh-A-DINO                      | 56.4 ± 2.0 | 49 ± 2.2   | 1.0 ± 0.0 |
> > | Oh-A-DINO (Hungarian Matching) | 55.6 ± 1.8 | 48.2 ± 2.0 | 1.8 ± 0.2 |
> >
> > * The results stay virtually the same when using Hungarian matching versus our matching approach. The explanation for this is in our opinion straightforward:
> >   * Hungarian matching is not necessary in our setting because the patch embeddings are highly discriminative: each query patch almost always selects a different candidate patch. Therefore, Hungarian matching becomes equivalent to our max-pool similarity and does not change the ordering or the scores.
> >   * This is empirically confirmed by our results on CLEVRTex and especially in the real-world results (see added visualisation for ImageNet), where correct material and texture retrieval would not be possible if multiple query patches collapsed onto the same candidate patch.
> >
> >
> > We hope this addresses all you concerns. Thank you!

---

> > > ### Comment · Reviewer_jkxw · 2025-11-25
> > > **Update 2**
> > >
> > > Thanks for the detailed response.
> > >
> > > But there is one more concern: I expected to see some challenging results from ImageNet, but the visualized image samples are all very simple. Are they from tiny-ImageNet? Specifically, images in Figure 8 and 9 are too simple, so could you provide some samples that are more complex than or at least as complex as those in Figure 7 the last two rows?

---

> ### Author Response · Authors · 2025-11-25
> **Response to Update 2**
>
> Dear Reviewer,
> * We agree that showing challenging ImageNet cases is useful. We chose the images in Figures 8 and 9 because they show a tradeoff between geometrical and surface-level complexity, while also being natural images from ImageNet.
>
> * **In addition to Figures 8 and 9, we now include a new set of ImageNet queries (Figure 11) containing multiple objects, clutter, occlusions, and diverse lighting.** These examples hopefully match the reviewer’s request and demonstrate that our method continues to retrieve images that preserve fine-grained color and material attributes under substantially more complex conditions.
>
> * Furthermore, since our approach uses DINOv2 as its backbone, it naturally inherits at least the same object-level retrieval capability. Our contribution is orthogonal: we improve the faithfulness of attribute retrieval, precisely the type of information SSL and slot-based representations often under-encode. Identifying such differences on ImageNet is inherently challenging because many classes do not exhibit controlled or systematically varying attributes, which is why finding illustrative examples is nontrivial. The newly added cases hopefully highlight the advantages of our method.

---

> ### Comment · Reviewer_jkxw · 2025-11-26
> **Update 3**
>
> Really appreciate the authors' responsive replies.
>
> But generally, I still doubt the effectiveness of the method to complex cases like ImageNet.
>
> > Identifying such differences on ImageNet is inherently challenging because many classes do not exhibit controlled or systematically varying attributes, which is why finding illustrative examples is nontrivial.
>
> Technically, for qualitative analyses or visualization, we have to ***pick*** some samples from the good results, which **say** account for 75% of the total results. But it is meaningless to ***pick*** the good results, which only account for 1% of the total -- The authors' case is, as the authors said, "*inherently challenging because ... nontrivial*". This indicates that the authors' claim about their method's effectiveness should be toned down appropriately.
>
> I decide to keep my original rating and leave the final decision to the collective judgment of other reviewers.

---

> > ### Author Response · Authors · 2025-11-26
> >
> > We would like to clarify one point regarding your final comment:
> > > “Technically, for qualitative analyses or visualization, we have to pick some samples from the good results … It is meaningless to pick the good results, which only account for 1% of the total.”
> >
> > * Our intention was not to selectively highlight rare successes. The ImageNet examples we provided were meant to demonstrate that our method also applies to more complex scenes, not to inflate its performance. Such cases are not isolated “1%” successes; they are reasonably common. The challenge lies instead in verifying attribute preservation on ImageNet, because (unlike CLEVR/CLEVRTex) ImageNet does not provide controlled attribute labels or consistent attribute variation. This makes quantitative evaluation inherently difficult, which is why our main claims rely on the structured datasets in the core experiments.
> >
> > * The central message of our work is unchanged and, we believe, appropriately stated:
> > we identify a systematic shortcoming in the standalone representations of existing object-centric and self-supervised models, and introduce a simple mechanism that faithfully restores surface-attribute structure.
> >
> > * This diagnostic contribution is supported by extensive controlled experiments in the main paper, and the ImageNet examples were added only to illustrate that the approach remains applicable in less controlled domains—not as the primary evidence.
> >
> > Thank you again for your engagement with our work.

---

### Author Response · Authors · 2025-11-19
**Summary of Rebuttal**

Dear Reviewers,

We want to sincerely thank you for your valuable feedback both positive and negative. We believe your feedback has led us to improve our paper in the revised version. Here is a quick summary of the changes that were made to address the concerns:

**1. Stronger OCL baselines.**
We added **SPOT** and **SmoothSA**, two state-of-the-art object-centric models, to all single- and multi-attribute retrieval evaluations. These models outperform SlotDiffusion on geometric cues but, importantly, still exhibit **consistently low colour/material retrieval**. This strengthens our core finding that OCL bottlenecks preserve surface cues only in reconstruction pathways, not in metrically accessible embeddings.

**2. Conceptual clarification.**
We clarified the difference between **information presence** and **geometric accessibility**, explaining why cosine similarity is appropriate for retrieval-based downstream tasks and how slot models can access surface-level features in VQA via supervised heads but that the standalone representations don't encode this information in a metrically accessible way. We updated the corresponding parts in the new revision of the paper.

**3. Clarified PCA inference and added examples.**
We clarified that PCA does **not** require multiple test inputs at inference. Instead, PCA directions are computed using a **fixed offline memory bank**, and each query retrieves _k_ nearest neighbours from this pool. Moreover, inference uses only a **single image** and our method requires only coarse foreground separation to work. We added segmentation examples in Figure 7 of the appendix. Finally, we clarify that any segmentation method could be used to retrieve a segmentation mask.

**4. Additional augmentation with DINO CLS token.**
Following suggestions, we evaluated SlotDiffusion **augmented with the DINOv2 CLS token**. This improves geometric attributes slightly but does **not** improve colour/material retrieval, confirming that our gains do not come from global features.

Furthermore, we attempted to address further concerns raised by each reviewer. We hope these additions and clarifications resolve the reviewers’ concerns and better communicate the contribution of our work.

---

> ### Author Response · Authors · 2025-11-24
> **New ImageNet Visualisations!**
>
> Dear Reviewers,
>
> Due to concerns voiced by reviewer jkxw, **we have added additional qualitative results in Figures 8, 9 and 10 of the appendix**. We show that our method is effective in settings with natural images and cluttered backgrounds, being able to retrieve material and colour more faithfully than DINOv2 and SlotDiffusion while remaining faithful to the geometric attributes.
>
> We hope this addresses further concerns you might have with our paper. Thank you!

---

### Author Response · Authors · 2025-12-02
**Discussion Summary and Final Notes**

Dear Area Chair,

Given the reassignment, we provide a brief summary clarifying the contribution and how the rebuttal addresses the reviewers’ concerns.
We want to sincerely thank you for your effort under these unusual circumstances.

---

## 1. Contribution and Scope

The paper identifies a **consistent representational limitation** in SSL and object-centric models:

* surface-level cues (color, material, fine appearance) are often present but **not metrically encoded** in object representations.
* this matters for downstream tasks that rely on **reasoning directly over representations**, for example retrieval, object selection, goal-conditioned RL, and similarity-based decision making without supervised decoders.
* our augmentation is simple and modular, improving the metric accessibility of these surface-level cues **without retraining the backbone**
* it also provides modest improvements on geometric attributes.

---

## 2. Two reviewers support acceptance

Reviewers **D46H** and **wQom** (both rating 6) clearly understood the scope and contribution.

* both gave positive evaluations.
* they agreed the limitation is real and relevant.
* they found the method clean and principled.
* they noted consistent improvements across CLEVR, CLEVRTex, Stanford Cars, and the ImageNet examples.

We fully resolved reviewer wQom’s remaining questions. Reviewer D46H had not yet responded when the discussion froze.

---

## 3. The rejecting reviews arise from a misunderstanding of scope, not methodological flaws

**Reviewer joPZ** evaluated the paper primarily through the lens of supervised-decoder tasks such as VQA and concluded that our setup “lacks motivation.”
This arises from a **fundamental scope mismatch**:

* supervised-decoder tasks learn attribute extraction in the prediction head, which can mask or override missing structure in the underlying representations.
* our contribution concerns the **geometric and metric structure of representations** in unsupervised or label-free downstream settings.

In the rebuttal, we clarified that:

* we do not claim information absence, only **geometric inaccessibility**.
* reconstruction ability does not imply metrically disentangled attributes.
* supervised-decoder tasks are therefore **not diagnostic** of the representational issue.
* cosine similarity is the standard diagnostic in DINO and CLIP style evaluation.

After addressing these points, we believe no methodological concerns remain.

**Reviewer jkxw** raised issues about baselines, PCA inference, matching, and ImageNet examples. We addressed all of these with additional experiments.

The reviewer’s final concern appears to be the qualitative difficulty of the ImageNet samples. Although ImageNet lacks attribute labels for quantitative evaluation, our controlled CLEVR, CLEVRTex, and Stanford Cars experiments demonstrate consistent robustness, while **Figures 8 to 11** provide qualitative confirmation that this robustness extends to natural images.

Nonetheless, the reviewer remarks that our ImageNet results represent only **“1 percent good cases”**, which is speculative and unsupported. The challenge lies instead in verifying attribute preservation on ImageNet, because (unlike CLEVR/CLEVRTex) ImageNet does not provide controlled attribute labels or consistent attribute variation for many of its classes. In any case, the ImageNet visualisations were added at the reviewer’s request and thus are supplementary, reinforcing the core evidence.

---

## 4. Strength of evidence

Across **all** tested SSL and OCL models (DINO, DINOv2-L, CLIP, SlotDiffusion, SPOT, SmoothSA):

* geometric cues are reliably encoded.
* color and material cues collapse in metric space.
* our augmentation of DINOv2 with object-level latents consistently resolves this issue.

---
Thank you again for your time and consideration and the time invested into assessing our paper!

---

### Meta-Review · Area_Chair_A2QC · 2026-01-05

**Summary:**

This paper makes an interesting observation: representations learned by object-centric representation learning methods exhibit different levels of "accessibility" via cosine similarity probes based on whether they encode shape/size or color/material/texture information. Shape/size being easily accessible with cosine similarity probes, whereas color/material/texture information is less well accessible. The authors train a separate model to learn an auxiliary latent space on extracted patches (centered on objects) which closes this gap.

The paper received two favorable reviews and two reviews that recommend rejection. One of the reject-voting reviewers highlighted in the existing discussion that they stick to their original score as they are still not fully satisfied with the paper.

The other reject-voting reviewer highlights concerns around relevance / scope of the paper: limiting the analysis to cosine similarity probes on relatively simplistic methods that have been optimized for unsupervised object segmentation performance (instead of linearly addressable latent spaces) seems artificial and not very relevant to the community overall. The authors defend against this argument highlighting that cosine similarity probes give a strong signal about usefulness of the method in standard retrieval pipelines. Nonetheless the AC finds this argument weak since the methods they are analyzing have clearly not been optimized for this task, and their main finding is in fact that they are not optimized for this setting. The proposed fix of learning a separate latent space on extracted patches adds little to the relevance or scope of the paper.

While the finding is interesting and the paper is otherwise very solid in terms of writing quality and experimental validation of the claims, the contribution is likely not of sufficient relevance for the broader community to warrant going against the recommendations by some of the expert reviewers who view this paper as not ready for publication at ICLR.

**Reviewer Concerns:**

Experimental validation and writing clarity concerns were successfully addressed, but the core issue around relevance/scope of the analysis remains unaddressed (as highlighted by reviewer joPZ).

**Reviewer Scores:**

The two reject-voting reviewers would have likely not changed their score (one of the two reviewers in fact highlighted in the discussion that they stick to their original score). The weak accept (score=6) voting reviewers may have increased their score as their concerns were mostly addressed but this would not have changed the fate of the paper.

---

### Decision · Program_Chairs · 2026-01-26

Reject